



# Organic Iron Complexes Enhance Iron Transport Capacity along Estuarine Salinity Gradients

Simon David Herzog[1], Per Persson[2], Kristina Kvashnina[3,4] and Emma Sofia Kritzberg[5]

[1] Department of Science and Environment, Roskilde University, DK-4000 Roskilde, Denmark
    [2] Centre for Environmental and Climate Research & Department of Biology, Lund University, SE-223 62, Lund, Sweden
    [3] The European Synchrotron, CS40220, 38043 Grenoble Cedex 9, France,
    [4] Helmholtz Zentrum Dresden-Rossendorf (HZDR), Institute of Resource Ecology, P.O. Box 510119, 01314, Dresden, Germany.
[5] Department of Biology/Aquatic Ecology, Lund University, SE-223 62, Lund, Sweden

*Correspondence to*: S. D. Herzog (sherzog@ruc.dk)

**Abstract.** Rivers discharge a notable amount of Fe ($1.5 \times 10^9$ mol yr$^{-1}$) to coastal waters, but are still not considered important sources of bioavailable Fe to open marine waters. The reason is that the vast majority of riverine Fe is considered to be lost

to the sediment due to aggregation during estuarine mixing. Recently however, several studies demonstrate relatively high stability of riverine Fe to salinity induced aggregation, and it has been proposed that organically complexed Fe (Fe-OM) can "survive" the salinity gradient, while Fe (oxy)hydroxides are prone to aggregation and selectively removed. In this study, we directly identified, by X-ray absorption spectroscopy, the occurrence of these two Fe phases across eight boreal rivers and confirmed a significant but variable contribution of Fe-OM in relation to Fe (oxy)hydroxides among river mouths. We

further found that that Fe-OM was more prevalent at high flow conditions in spring than at low flow conditions during autumn, and that Fe-OM was more dominant in low-order streams in a catchment than at the river mouth. The stability of Fe to increasing salinity correlated well to the relative contribution of Fe-OM, i.e. confirming that organic complexes promote Fe transport capacity. This study suggests that boreal rivers may provide significant amounts of potentially bioavailable Fe to marine waters beyond the estuary, due to organic matter complexes.






## 1 Introduction

Iron (Fe) mobility from the litho- and pedosphere into the hydro- and biosphere is controlled by physical, chemical, and biological processes. While Fe is the fourth most abundant element on earth (Taylor, 1964), Fe concentrations in oxygenated aquatic system are generally low (Johnson et al., 1997;Kraemer, 2004), while they can be higher during high flow conditions

and in boreal waters with high dissolved organic carbon concentrations (DOC) (Kritzberg et al., 2014;Ekström et al., 2016). The more soluble form of Fe - Fe(II) - is favored under strongly reducing or highly acidic conditions (Waychunas et al., 2005). At circumneutral pH and oxic conditions Fe(II) gets oxidized to Fe(III), which has a low solubility and precipitates as Fe (oxy)hydroxides (Lofts et al., 2008). Thus, mobilization of Fe to surface waters requires either water flow through anoxic soil layers, favoring Fe(II), or that Fe is complexed by organic ligands and becomes mobile also in oxic soil layers (Tipping,

1981;Stumm and Morgan, 1970).

Several studies report rising Fe concentrations in surface waters, especially in Northern Europe (Neal et al., 2008;Kritzberg and Ekstrom, 2012;Sarkkola et al., 2013;Weyhenmeyer et al., 2014;Björnerås et al., 2017), suggesting that Fe export from soils are increasing. As a consequence, Fe loading from boreal rivers to estuaries is increasing substantially (Kritzberg and Ekstrom, 2012;Kritzberg et al., 2014;Björnerås et al., 2017). Given the key role that Fe plays in both local and global

biogeochemical cycles in coastal and marine systems, this is a finding with major implications to the receiving systems. What the consequences may be, depend first and foremost on the fate of Fe in the estuarine salinity gradient. To the extent that Fe is stable to salinity induced aggregation and sedimentation, it may provide potentially bioavailable Fe to the marine system. However, Fe is known to behave non-conservatively in estuaries, and it has been suggested that at least 95 % of Fe is aggregated and lost to the sediments in the early part of estuarine mixing (Sholkovitz et al., 1978;Haese, 2006). Fe can

play an important role in the sediment *e.g.* by acting as a C and P sink (Lalonde et al., 2012;Lenstra et al., 2018).

Fe transport capacity – the fraction of riverine Fe remaining in suspension at higher salinity – has been shown to vary widely and is in some instances much higher than generally observed (Kritzberg et al., 2014;Krachler et al., 2005). Thus the riverine Fe source to marine waters may be underestimated, especially for boreal rivers, where high DOC concentrations can affect Fe speciation. Fe in natural waters is known to occur in two main phases, mononuclear organic complexes (Fe-OM) and Fe-

rich Fe (oxy)hydroxide colloids associated with chromophoric organic matter (Breitbarth et al., 2010;Hassellöv et al., 1999;Andersson et al., 2006). It has been suggested that variability in Fe transport capacity between rivers (1% to 55%) may be explained by a varying proportion of these Fe phases (Kritzberg et al., 2014). However, the postulated link between Fe-OM and Fe transport capacity requires direct assessment of Fe speciation to verify previous interpretations based on Fe:OC ratios and molecular size (Stolpe and HasHassellöv 2007; Krachler et al. 2010; Kritzberg et al. 2014). A preferential loss of

Fe (oxy)hydroxides by aggregation was shown by Herzog et al. (2017) using X-ray absorbance spectroscopy. While this infers that Fe-OM is more stable to salinity induced aggregation, some Fe-OM was also found aggregates, indicating that the control of Fe stability is more complex.

Based on the previous findings, the aim of the current study was to better understand what controls the fate of Fe from boreal rivers across estuarine salinity gradients by exploring 1) if variability in relative contribution of Fe-OM and Fe

(oxy)hydroxides can explain variation in Fe transport capacity, and 2) if the relative contribution of Fe-OM and Fe (oxy)hydroxide is controlled by spatial factors and flow conditions, within and among rivers.

To this purpose, we sampled eight river mouths, chosen to encompass a wide geographical and climatic gradient, as well as a range in Fe and DOC concentrations. The Fe speciation of all river samples was characterized by XAS. To be able to link Fe speciation to Fe transport capacity, the same river waters were exposed to artificial salinity gradients (mixing experiments).

Four of the rivers were sampled under high (spring) and low flow conditions (autumn). To reveal differences along the flow path, two upstream sites were sampled in addition to the river mouth in one river catchment.



## 2 Materials and Methods

### 2.1 Site description and sampling

Eight rivers around the Swedish coast with distinct differences in climate and catchment characteristics (Figure 1) were
selected for this study. The annual temperature ranged from 5.8°C for the most northern (Öre) and 9.8°C for the most
southern river mouth (Helge). Forest is the most dominant land cover and peat soils are present to a varying extent in all the
catchments (Table 1). While six of these river mouths were sampled for previous studies exploring Fe dynamics in response
to increasing salinity (Kritzberg et al., 2014;Herzog et al., 2017), relating the XAS assessed contribution of Fe-OM
complexes to Fe transport capacity, was not previously done. Moreover, to investigate differences in Fe speciation along a
river, two upstream locations in river Helge were included. The most upstream sample site, Svineö, is draining from a peat
bog, with a high percentage of peat soil. The second site along the river path, Biveröd, is a small $2^{nd}$-order stream in a
predominantly forested landscape. In addition, to understand the impact of high and low flow conditions on Fe speciation
and transport capacity, sampling were carried out twice – during autumn and spring – in four of the river mouths (Emån,
Lyckby, Mörrum and Helge). A major difference in the discharge between the autumn and spring sampling was observed
(Table 1). Finally, for two rivers (Öre and Örekil), transects were sampled starting at the river mouth and extending over the
estuarine salinity gradient, to facilitate comparison of Fe transport capacity by mixing experiments and *in situ* Fe
concentrations along the natural salinity gradient.

Water was sampled by hand from half a meter below the surface into acid cleaned polyethylene containers through a 150-µm
nylon mesh. The mesh was used to ensure homogeneous samples free of large detritus. No further filtration steps were
applied, to ensure that all suspended Fe was included, which is critical when studying the stability of riverine Fe across
salinity gradients. Samples were stored cold and dark until return to the lab.

For the XAS analysis, a 1-L sample of water was frozen as soon as possible and never more than 5 hours after collection.
The samples were later freeze-dried and stored dry in the dark until analyzed.  Freeze-drying is commonly applied for
preservation and pre-concentration of XAS samples (Karlsson, et al. 2008; Vilgé-Ritter, et al. 1999). While the freezing may
lead to changes in the physical properties of colloids (Raiswell, et al. 2010), the chemical composition should be less
affected. For instance, drying/freeze-drying could increase the rate of crystallization of Fe (Bordas and Bourg 1998),
reducing the amorphous Fe phase compared with native samples, but this would not affect the distinction between
organically complexed Fe and Fe (oxy)hydroxides which was the focus here.

Oxygen and pH were measured in situ with OxyGuard MkIII and SevenGo Duo pH meter (Mettler Toledo), respectively.
Only acid washed material was used for sample handling, and for XAS and total Fe measurement polycarbonate
bottles/containers were used.

### 2.2 Artificial Seawater Mixing Experiments

Experiments mixing river water and artificial seawater were initiated as soon as possible, and no later than 3 hours after
sampling. Water samples were mixed with artificial sea salt solutions in a 6:1 ratio (vol:vol) in 50-ml Falcon tubes to 8 or 16
levels of salinity ranging from 0 to 35. To achieve the desired final salinities the added concentration of the sea salt solution
varied. These were made from an artificial sea salt stock solution produced using reagent grade salts (Sigma Aldrich)
following a standard protocol (Kester et al., 1967) (mass fraction given in %: $Cl^-$ (55.05), $Na^+$ (30.62), $SO_4^{2-}$ (7.68), $Mg^{2+}$
(3.69), $Ca^{2+}$ (1.15), $K^+$ (1.10), $HCO_3^-$ (0.40), $Br^-$ (0.19), $H_3BO_3$ (0.07), $Sr^{2+}$ (0.04), $F^-$ (0.003)). The Fe contamination from
the salts used was negligible, e.g. the addition of salt to produce salinity 35 added a maximum of 0.15 µM Fe, which
corresponds to 0.1-3.8 % of the Fe concentration in the river waters studied. The stock solution was diluted to the desired
concentration by Milli-Q water (Millipore, 18.2 MΩ). After mixing the river water with the salt solution, the samples were
kept in the dark on a shaker for at least 24 h to allow aggregation. Salinity induced aggregation of Fe consists of several





reactions with a significant fraction aggregating within a few seconds (Nowostawska et al., 2008). While aggregation then continues at a slower rate after the first few hours (Nowostawska et al., 2008;Hunter and Leonard, 1988), the first 24 h
should include the largest fraction of the Fe removal.

The aggregates were separated by centrifugation at 3000 rcf for 8 h at 4°C. After centrifugation Fe, DOC, pH and salinity were measured in the supernatant. Fe transport capacity was calculated as the Fe concentration in the supernatant divided by the *in situ* Fe concentration and multiplied by 100 (%).

**2.3 Standard Analytical Methods**

An ICP-AES Optima 3000DV (Perkin Elmer) was used to determine total Fe concentration on acidified samples (1% vol, $HNO_3$). A Shimadzu TOC V-CPN was used to analyze organic carbon by high temperature catalytic-oxidation, using the Non-purgeable Organic Carbon (NPOC) method. For calibration a five-point standard curve was used and blanks and standards were included in all runs. The pH of the mixing experiment samples was measured with a 913 pH Meter (Metrohm) and salinity was determined by a WTW inoLab Cond730.

**2.4 XAS Data Collection and Analysis**

Synchrotron data was collected at the beam line I811 MaxII ring; Max Lab, Lund University. Fe K-edge XAS spectra were collected on the river samples at fluorescence mode at room temperature. A Lytle detector with an Mn-filter (3µx) was used to minimize unwanted scattering and fluorescence contributions. Aligned samples at 45° relative to the incident beam guaranteed an optimal fluorescence signal. Depending on the Fe concentration 15 to 40 spectra for each samples were
recoded taking approximately five minutes each. Transmission scans of a reference Fe foil were collected simultaneously during all scans for energy calibration.

Data treatment and analysis for the extended X-ray absorption fine structure (EXAFS) and wavelet transform (WT) spectra were according to Herzog et al. (2017). A subset of the samples (6 out of 14) was previously analyzed for EXAFS (Herzog et al., 2017), and reanalysis was performed with consistent fitting parameters for the whole data set to allow comparison
among all samples. In short, all scans were checked for beam damage before averaged with SixPack (Webb, 2005). The averaged scans were normalized and the background was removed by subtracting a spline function in Viper (Klementiev, 2000). The same program was used for investigation of self-absorbance and shell by shell fitting of the EXAFS data. For the WT analysis Igor Pro script was used (Funke et al., 2005). The $k^3$-weighted spectra were modeled in k-space from 2.8-12.0 $Å^{-1}$ using theoretical phase and amplitude functions from FEFF7 (Zabinsky et al., 1995). Goethite (O'day et al., 2004) and
the trisoxalatoiron(III) complex (Persson and Axe, 2005) were used as input structures for calculations with FEFF. While fitting, the amplitude reduction factor ($S_0^2$) was set to 0.70. Further, with values found in the literature, the numbers of free variables were restriction by correlating coordination numbers and fixing the Debye–Waller factors ($\sigma^2$). Also, the threshold energy ($\Delta E_0$) beyond the first was assumed to be identical for all shells.

A linear combination fitting (LCF) analysis was applied to the river mouth samples using SixPack. $k^3$-weighted EXAFS
spectra (k 3.0 to 12.0 $Å^{-1}$) were used in the LCF analysis. Reference spectra of ferrihydrite, goethite, hematite, lepidocrocite and a Fe(III) complexed to Suwannee Rives fulvic acid were used as model compounds. This model provided good fits for all samples and further allowed us to distinguish between organically complexed Fe and Fe (oxy)hydroxide. During the LCF analysis $E_0$ was allowed to float, a non-negative boundary condition was applied and the sum of species was not forced to equal 100%. Components with a contribution less than 5% were excluded from the models.

Fe K-edge high-energy resolution fluorescence detection (HERFD) XANES spectra and Fe $Kb_{2,5}$ emission spectra were collected at the high-brilliance X-ray absorption and X-ray emission spectroscopy undulator beamline ID26 of the European Synchrotron Radiation Facility (ESRF, Grenoble) (Signorato et al., 1999). The incident X-ray beam was monochromatized with a pair of cryogenically cooled Si(111) crystals. The sample, analyzer crystal, and photon detector (silicon drift diode)





were arranged in a vertical Rowland geometry. The intensity was normalized to the incident flux. Both the HERFD and the
       Fe $Kb_{2,5}$ emission spectra were measured using the X-ray emission spectrometer (Glatzel and Bergmann, 2005;Kvashnina
       and Scheinost, 2016), and the HERFD measurements were performed by recording the intensity of the Fe $Kb_1$ emission line
       (7058 eV) as a function of the incident energy using five Ge (620) crystal analyzers at 79° Bragg angle.

### 2.5 Data treatment

       The contribution of the single Fe-C and the multiple Fe-O/C scattering paths, was statistically analyzed in the EXAFS fits by
an F-test with Viper (Klementev, 2001).
       As a measure of the relative contribution of Fe-OM and Fe (oxy)hydroxides in the water samples, two different approaches
       using the $k^3$-weighted EXAFS spectra were applied: 1) a ratio of the coordination numbers of the fitting results, between the
       Fe-C path and the shortest (edge-sharing) Fe-Fe path (i.e. $CN_{Fe-C}/CN_{Fe-Fe}$); 2) a ratio of the Fe-OM fraction and the sum of
       Fe-oxide fractions from the LCF analysis.
Relationships for the river mouth samples between Fe transport capacity at 35 salinity (corresponding to the salinity of the
       open sea), $CN_{Fe-C}/CN_{Fe-Fe}$ ratios, LCF ratio, Fe:OC ratios, total Fe, and DOC were tested by Pearson correlations.
       Assumptions of normality were verified by Shapiro-Wilk tests. Differences in Fe transport capacity between spring and
       autumn samples, as well as DOC concentrations at *in situ* and 35 salinity in the mixing experiment, were tested by paired t-
       tests.
Expected, or theoretical, values of *in situ* Fe across the estuarine salinity gradient were calculated by accounting for the
       dilution of riverine water by sea water (estimated by salinity) and the stability of Fe to aggregation as assessed by the
       artificial mixing experiments. The following equation Eq. (1) was used to calculate the expected Fe concentration [$Fe_{exp}$] at a
       given salinity:

$$[Fe_{exp}] = \frac{Fe_{river} \times salinity}{Fe_{stability} \times salinity_{marine\ end-member}} + [Fe_{marine\ end-member}] \times \left(1 - \frac{salinity}{salinity_{marine\ end-member}}\right) , \qquad (1)$$

where [$Fe_{river}$] is the Fe concentration at the river mouth, $salinity_{marine\ end-member}$ is the highest salinity in the estuarine *in situ*
       gradient, [$Fe_{marine\ end-member}$] is the Fe concentration at the highest salinity in the estuarine gradient, and $Fe_{stability}$ is the fraction
       that remained in suspension in the artificial sea water mixing experiments at the given salinity.

### 3 Results

#### 3.1 Water chemistry

At the time of sampling all river mouths were close to saturated with dissolved oxygen (85 – 118%) and pH values close to
       neutral (Table 2). For the river mouths that were sampled twice, pH was consistently lower during spring than during autumn
       sampling, i.e. lower during higher discharge. Total Fe concentrations in the river mouths varied from 0.22 to 2.28 mg/l and
       DOC concentrations from 8.8 to 24.2 mg/l. Water chemistry in the two upstream samples from the Helge catchment differed
       strongly from those of the river mouth. Oxygen saturation and pH were lower in the upstream sites, especially in Svineö
(dissolved oxygen saturation 41% and pH 4.4), while Fe and DOC concentrations were markedly higher. Across all samples
       Fe and DOC concentrations were strongly correlated (r = 0.96; p <0.001), but since Fe was more variable (32-fold) than
       DOC (6-fold) there was a wide range in Fe:DOC molar ratios, from 0.005 to 0.035.

#### 3.2 XAS characterization

       The XAS analyses identified two main Fe phases, namely Fe (oxy)hydroxide, and Fe ions associated with organic matter as
Fe-OM complexes. Both phases were qualitatively identified in the WT contour plots (Figure 2). The feature in the WT plots
       at ca. 7.5 $Å^{-1}$, 2.8 Å, originate from Fe-Fe scattering paths (denoted in the first plot in Figure 2 as Fe) and are similar to that





of ferrihydrite (Sundman et al., 2014;Yu et al., 2015) and goethite (Karlsson and Persson, 2010;Sundman et al., 2014). Features for the Fe-OM complexes occurred at ca. 3 Å$^{-1}$, 2.5 Å and 3 Å$^{-1}$, 3.2-3.7 Å, caused by single Fe-C and multiple Fe-C-C(O or N) scattering, respectively (denoted in the first plot in Figure 2 as C and C/O) and are in good agreement with

previously identified Fe-OM complexes (Karlsson and Persson, 2010). In Figure 2 only a selection of the WT plots are shown, all WT plots of the remaining samples can be found in Figure S1.

Guided by the WT results, the EXAFS spectra were quantitatively modeled by a shell-by-shell non-linear least squares fitting procedure (Table S2) including five paths (Fe-O, two Fe-Fe, Fe-C and a Fe-C/O multiple scattering path). The two Fe-Fe paths were used to describe the contribution of Fe (oxy)hydroxide and the Fe-C and the Fe-C/O multiple scattering

paths were used for the Fe-OM component. This modeling approach corroborated the qualitative WT analyses and provided good fits to all spectra (Figure S2). For the river mouth samples the Fe-Fe edge- and corner-sharing distances were determined to 3.05-3.11 Å and 3.41-3.46 Å, respectively (Table S1). The coordination number (CN) of the short Fe-Fe path varied between 1.0 and 2.7, indicating significant contribution from Fe (oxy)hydroxide, as corroborated by the WT plots. The Fe-C distances varied between 2.85 and 3.00 Å and the CN of the Fe-C path varied between 0.9 and 2.8 Å. To verify the

contribution of the Fe-C path, an F-test comparing EXAFS models with and without the Fe-C and the Fe-C/O multiple scattering paths was performed, showing a significant contribution at the 92 % confidence level or better. While the EXAFS fitting analyses confirmed the presence of both Fe (oxy)hydroxide and Fe-OM complexes in all river mouth samples, the CN values indicated a large variation in the relative contribution of these Fe species (Table S1).

LCF analysis supported the variable contribution of the two Fe phases in the water samples (Table S2) and was in good

agreement with the WT data. LCF analysis assigned the main components in the river mouth samples to Fe-OM, ferrihydrite and lepidocrocite. The contributions from goethite and hematite were below 5% in all samples and therefore excluded from the final analysis.

Differences in the relative contribution of Fe-OM and Fe (oxy)hydroxide across the samples were obvious from the WT plots (Figure 2 and Figure S1) and supported by the $CN_{Fe-C}/CN_{Fe-Fe}$ and LCF ratios (Table 2). There was agreement between

the two ratios as indicated by correlations across all samples (r = 0.58; p = 0.047) and river mouth samples only (r = 0.64; p = 0.033). Low ratios, i.e. low contributions of Fe-OM, were indicated by both approaches for Rivers Öre, Alster, Lyckeby$_{autumn}$, and Örekil, for which the Fe (oxy)hydroxide feature was dominating in the WT plots. This was contrasted by River Lyckeby$_{spring}$ and Mörrum$_{spring}$, with high ratios and particularly strong Fe-OM signals in the WT plots. For the other river mouth samples a more even contribution of the two Fe species was found.

Fe speciation in the two samples taken upstream in the River Helge catchment was distinctly different from the river mouth. The HERFD and Kb$_{2,5}$ emission spectra indicated a gradual change from low to high Fe oxidation state when approaching the river mouth. In the HERFD spectra the presence of Fe(II) was shown by a low energy shoulder at 7.1135 kEv, which was most pronounced for the Svineö sample (Figure 3). This effect was even more obvious in the Kb$_{2,5}$ emission spectra where the peak at 7.105 kEv provided direct evidence for the presence of Fe(II), based on comparison with the model compounds.

The emission spectrum of the Svineö sample is very similar to the organic Fe(II) complex Fe(acac)$_2$, which suggests that Fe(II) was the predominant oxidation state at this site.

The WT results of the Svineö sample indicated no Fe-Fe scattering (Figure 4), whereas further downstream at Biveröd both Fe-C and Fe-C-C(O or N) features were present. In the river mouth Fe-Fe was even further pronounced. Thus, the Fe-Fe signal increased in strength the further downstream the sample was taken.

The LCF analysis corroborated the WT results; i.e. for Svineö no Fe-oxides were identified, whereas in Biveröd both Fe-OM and Fe-oxides were present and in the Helge river mouth Fe-oxides were dominating (Table S2). This trend was also obvious from the quantitative shell-by-shell fitting results (Table S1). Finally, comparing the various EXAFS analyses with the HERFED and Kb$_{2,5}$ emission spectroscopy results show that Fe(II) in the Helge river system is present as Fe-OM complexes. These complexes are favored by low pH and low oxygen concentrations, as expected.



For the river mouths that were sampled twice, samples taken during high flow regime in spring displayed consistently higher $CN_{Fe-C}/CN_{Fe-Fe}$ and LCF ratios than samples taken in the same river in autumn during lower discharge, indicating a higher contribution of Fe-OM complexes during high flow conditions (Table S1).

Considering all samples, the relative contribution of Fe-OM and Fe (oxy)hydroxide was not predicted by the molar Fe:OC, i.e. no correlation between the Fe:OC and the Fe speciation ratios was found ($CN_{Fe-C}/CN_{Fe-Fe}$ ratio: r = 0.182, p = 0.570; LCF

ratio: r = 0.53; p = 0.077). Instead, the LCF ratio was negatively correlated to pH (r = -0.69; p = 0.019). Among all samples, no significant relationships were found between $CN_{Fe-C}/CN_{Fe-Fe}$ or LCF ratios and variables related to catchment size, land cover or soil type.

### 3.3 Fe transport capacity.

The general pattern of the artificial seawater mixing experiments was a non-conservative behavior of Fe with increasing

salinity (Figure 5). Fe removal took place already at low salinities, with more than 50% of the Fe removed at salinity 2 for some river samples (Öre, Alster, Mörrum_autumn and Helge_spring, Helge_autumn). At a salinity corresponding to the open ocean (35) between 76 and 93% of Fe was removed. High Fe transport capacity was measured for river Lyckeby with 24% remaining in suspension at salinity 35. No significant loss of OC in response to increasing salinity was found for rivers where OC was analyzed in the mixing experiment (Öre, Örekil, Helge, Mörrum, Emån and Lyckeby; $t_5 = 1.38$ p = 0.17).

For the river mouth samples, the Fe transport capacity at 35 salinity correlated positively with the Fe speciation ratios ($CN_{Fe-C}/CN_{Fe-Fe}$: r = 0.675, p = 0.023; LCF ratio: 0.78, p = 0.005). Further, Fe transport capacity at 35 salinity were negatively correlated to pH (r = -0.730, p = 0.007).

When comparing the samples taken within the Helge catchment, upstream samples showed a higher Fe transport capacity than the river mouth sample (Figure 5). For the four river mouths that were sampled twice, Fe transport capacity (at 35) was

consistently higher in spring than in autumn ($t_3 = 4.696$, p = 0.0183).

In addition to raising the salinity, also the pH in the samples increased due to the experimental treatments from 6.4 - 7.5 in the river mouths to 7.4 - 8.3 in the high-salinity treatments, and for the upstream samples from pH 4.7 - 6.5 to pH 7.1 - 7.7.

*In situ* concentrations of total Fe declined with increasing salinity and distance from the river mouths (Figure S3). The theoretical values for Fe, calculated based on salinity induced aggregation in the artificial seawater mixing experiments and

dilution estimated by salinity, were only slightly lower than the measured *in situ* values for River Örekil. The deviation was larger for River Öre, where the Fe measured *in situ* was substantially higher than the theoretical values at the lower levels of salinity.

## 4 Discussion

### 4.1 In situ speciation of Fe

Two carrier phases associated with Fe transport in freshwater in boreal system - Fe-OM and Fe (oxy)hydroxides (Andersson et al., 2006;Hassellöv et al., 1999) - have been previously verified by XAS (Sundman et al., 2013;Herzog et al., 2017) and were also identified and detected in all river mouth samples in this study. The results of the quantitative modeling of the EXAFS spectra and LCF analysis correlated, which is satisfying considering the potential sources of error of both analysis. Results showed a wide variation in the relative contribution of the Fe phases across river mouths, with some dominated by

Fe-OM, *e.g.* River Lyckeby_spring, and some by Fe (oxy)hydroxide, *e.g.* River Öre. The significant contribution of Fe-OM in the river mouth samples is in contrast to thermodynamic modeling, which has suggested a dominance of ferrihydrite (≥97 %) for these systems (Wällstedt et al., 2010).

The contribution of the Fe phases changed markedly along the Helge river catchment. In the water draining from a peat bog (Svineö), Fe(II) was predominant and only Fe-OM was present. Neither EXAFS fitting nor LCF identified Fe-Fe paths to





support the presence of Fe (oxy)hydroxide. The sample showed similar features as groundwater in a northern boreal catchment (Sundman et al., 2014) or Fe(II) loaded onto peat humic acid (Yu et al., 2015). Further downstream (Biveröd), a mixture of the two phases could be seen, similar to samples from soil waters from organic layers close to a boreal stream (Sundman et al., 2014). The Fe(III) and Fe (oxy)hydroxide fraction was more prominent along the river path and was highest at the river mouth.

In addition to variation across river mouths and within the Helge river catchment, XAS analyses revealed a clear variation between high and low flow regime in the river mouth samples. Samples collected during high flow in spring showed a higher contribution of Fe-OM than the autumn samples, which is a direct support for previous interpretations made by ultra filtration (Pokrovsky et al., 2010;Stolpe et al., 2013). This variation is likely to be driven by changes in the source of the Fe depending on seasonal dynamics in hydrology (Dahlqvist et al., 2007).

**4.2 Fe transport capacity and the link to Fe speciation**

Non-conservative behavior of Fe was seen in the artificial mixing experiments as well as in the estuarine gradients sampled, and is consistent with existing literature (Gustafsson et al., 2000;Boyle et al., 1977;Sholkovitz, 1976). Fe transport capacity varied from 0.7 to 24 % among the rivers. The high FeTC for most of the rivers studied go along with field and laboratory studies showing that high-latitude DOC-rich rivers exhibit higher Fe-carrying capacities (Sholkovitz, 1976;Powell et al.,
1996;Krachler et al., 2005). DOC was little affected by increasing salinity as previously observed (Linkhorst et al., 2017;Herzog et al., 2017).

Previous studies using size separation and spectrometric methods (Stolpe and Hassellöv, 2007), but also XAS (Herzog et al., 2017), have suggested that mainly Fe (oxy)hydroxide is affected by salinity and selectively lost from suspension by aggregation and sedimentation (Herzog et al., 2017). In contrast, Fe complexed by terrigenous organic matter is supposedly
"surviving" estuarine mixing and can thereby be a source of bioavailable Fe to marine waters (Krachler et al., 2010;Laglera and van den Berg, 2009;Batchelli et al., 2010). The positive correlation between the contribution of Fe-OM (as determined by XAS) and Fe transport capacity (determined in artificial mixing experiments) adds a direct support that organic complexation of Fe is enhancing the stability across salinity gradients.

In freshwater Fe (oxy)hydroxide is stabilized by surface interactions with organic matter (OM), providing a negative surface
charge (Sander et al., 2004;Tiller and O'Melia, 1993). With increasing salinity, the surface charge gets neutralized resulting in reduced colloidal repulsion (Sander et al., 2004;Mosley et al., 2003). Marine cations, like magnesium and calcium, which neutralize the negatively charged surface groups of the OM, weaken the interaction between colloidal Fe (oxy)hydroxide and OM (Turner and Millward, 2002) and further promote the destabilization of Fe (oxy)hydroxides at increasing salinity. The same cations may favor release and hydrolysis of Fe-OM, by competing for the binding sites of the organic ligands (Fujii et
al., 2008). The stability of organically complexed Fe may also be reduced at high ionic strength leading to compression of water and consequent "salting out" of the organic complexes (Turner and Millward, 2002;Turner et al., 2004). Accordingly, Herzog et al. (2017) also identified organically complexed Fe in salinity-induced aggregates especially at high salinities. However, both the selective loss of Fe (oxy)hydroxide at saline conditions, and the positive correlation between the relative contribution of Fe-OM and transport capacity in the current study underpin the role of Fe speciation in controlling the fate of
Fe across salinity gradients. It would be an advantage to directly measure Fe speciation remaining in suspension in saline samples, to see if Fe (oxy)hydroxide is present, but this is currently hindered by methodological limitations. It has been suggested that Fe isotopic ratios may reflect Fe speciation (Ingri et al., 2006;Ilina et al., 2013), however this remains to be confirmed.

Results regarding Fe transport capacity derived from the artificial seawater mixing experiments were in good agreement with
the estuarine transects sampled. Theoretically calculated Fe concentrations, based on Fe loss in artificial seawater mixing experiments with river water and the dilution factor, showed only minor deviations from Fe concentrations measured in the



Gullmar Fjord. For the Öre estuary on the other hand, measured Fe concentrations were somewhat higher than the theoretical calculations (Figure S3). In the low-salinity mixing regime present here, aggregation may occur without significant sedimentation (Forsgren and Jansson, 1992). This has been observed in the plume of nearby River Kalix, and was

hypothesized to result from a high organic component of the aggregates, where low specific density may lead to transport of these aggregates far away from the river mouth (Gustafsson et al., 2000). Thus the centrifugation used to efficiently separate aggregates in the mixing experiments, may overestimate estuarine particle loss in this context.

**4.3 Control of spatial variation and flow conditions on Fe speciation.**

The results of this study showed that Fe speciation is highly variable across spatial scales and during different flow

conditions and is further linked to Fe stability across salinity gradients. It is therefore imperative to understand what factors govern Fe speciation. The largest variability in Fe speciation was observed between samples taken along the flow path of River Helge. In the most upstream location, which drain a major peat bog (Svineö), Fe(II) and Fe-OM dominated. This site also showed low oxygen and pH but high DOC concentration – conditions that favor complexation over Fe(III) hydrolysis (Neubauer et al., 2013). The Fe speciation in the stream water close to the Fe sources is thus determined by the properties of

the in-flowing terrestrial Fe(II)/Fe(III)-OM complexes and of the conditions when anoxic, hydrated Fe(II) meets oxygenated DOC-rich waters (Sundman et al., 2014). The contribution of Fe(III) and Fe (oxy)hydroxide increased with pH and oxygen saturation along the flow path. This is in agreement with Neubauer et al. (2013), who argued that pH and OM were the main factors controlling Fe speciation in a boreal catchment, and explaining the dominance of Fe-OM in wetland-influenced headwaters and increasing Fe (oxy)hydroxide downstream (based on molecular size and chemical equilibrium modeling).

The difference in speciation along the flow-path may in part be due to organically complexed Fe precipitating as Fe (oxy)hydroxide due to strong hydrolytic tendencies (Karlsson and Persson, 2012) as pH increases and OM declines, and due to photo reduction of Fe(III)-OM (Fujii et al., 2011;Waite and Morel, 1984;Neubauer et al., 2013). But the difference may also reflect different sources of Fe to the river along the flow-path. Thus while runoff from organic soil layers may bring predominantly Fe-OM to low order streams (Lydersen et al., 2002;Abesser et al., 2006;Dahlqvist et al., 2007;Sundman et al.,

2014), groundwater inflow is more significant further downstream and may bring small Fe (oxy)hydroxides or Fe(II), which is rapidly hydrolyzed in the riparian zone when anoxic groundwater and oxic waters mix (Vasyukova et al., 2010). Hence, the speciation of Fe at the river mouth is determined both by the properties of the inflowing water and the chemical processing along the river flow path.

A consistent pattern was that samples taken in spring, when discharge was higher, showed a larger contribution of Fe-OM

than autumn samples from the same river mouths. Temporal variation in river runoff is tightly linked to different hydro-geological pathways (Andersson et al., 2006;Pokrovsky et al., 2006;Neff et al., 2006). During autumn, dominance of groundwater input, and longer residence time of ground water, should promote input of Fe(II), which rapidly oxidizes to form Fe (oxy)hydroxides in surface water in the absence of high OM concentrations (Dahlqvist et al., 2007). During high discharge on the other hand, like during spring flood or high precipitation events, organically complexed Fe gets mobilized

from the upper soil layers into the river, due to raising water tables and surface runoff (Grabs et al., 2012;Dahlqvist et al., 2007). The lower pH and higher DOC values in our spring samples agrees with this reasoning – that during higher discharge flow through organic-rich soil layers has a higher influence on river water chemistry and allows for the formation of more Fe-OM.

It was notable how Fe speciation and Fe stability matched pH across the entire dataset. A high contribution of Fe-OM and

high stability coincided with low pH, across river mouths, along the flow path of the Helge catchment, and in the spring samples compared to the autumn samples. pH should exert a strong control on Fe speciation and increasing pH favors precipitation of Fe (oxy)hydroxide due to the strong hydrolytic tendency of Fe(III) (Karlsson and Persson, 2012). Moreover, a low pH may reflect strong influence from organic soils where Fe prevails as Fe-OM on water chemistry, as seen in low



order systems and under high flow conditions (Dahlqvist et al., 2007;Neubauer et al., 2013). On the contrary, a high pH in the river mouth in these boreal systems may reflect a higher influence of groundwater input further downstream, possibly increasing the contribution of Fe (oxy)hydroxide. The increasing pH downstream in parallel with photoreduction and declining DOC concentration, may lower the stability by promoting release and hydrolysis of organically complexed Fe originating from organic soils further upstream (Neubauer et al., 2013;Waite and Morel, 1984;Fujii et al., 2011).

The temporal variability within rivers suggests that Fe speciation at a given time is not well predicted by catchment characteristics only. While characteristics such as land-cover and soil type are most likely affecting both amounts and speciation of Fe exported from the catchment, the limited number of rivers and sampling occasions of this study cannot accurately discern such relationships.

**5 Conclusions, implications and future perspectives**

The collective results from this study confirmed the existence and wide variability in the contribution of two Fe phases – Fe-OM complexes and Fe (oxy)hydroxides – among the rivers included. It further confirmed that the response of river-borne Fe to increasing salinity differed widely. Interestingly, the differences in stability towards salinity-induced aggregation matched well the differences in relative contribution of Fe-OM across the river mouths, between high and low flow conditions, and along the flow path of a river catchment. Thus, by assessing the Fe speciation by XAS, this study provides direct evidence that Fe-OM enhances survival over estuarine salinity gradients.

This would suggest that high and rising concentrations of Fe from boreal rivers (Kritzberg and Ekstrom, 2012;Björnerås et al., 2017) may indeed result in increasing export of bioavailable Fe to open waters, where it is frequently limiting N-fixation and primary production (Stal et al., 1999;Stolte et al., 2006;Martin and Fitzwater, 1988). Major hydrological events like spring floods and heavy storms have been observed to increase of the Fe concentration by up to a factor of 20 and alter the annual Fe load in northern rivers (Hölemann et al., 2005;Rember and Trefry, 2004). The hydro-geological conditions during such events (Dahlqvist et al., 2007), may promote a higher contribution of Fe-OM and thus a higher stability during estuarine mixing, resulting an increase of Fe export into the open waters. However, sampling with higher temporal resolution would be required to substantiate such an extrapolation. Moreover, since the majority of the Fe does aggregate in response to increasing salinity, the increases in Fe discharge is also likely to alter *e.g.* P retention in coastal sediments (Rosenberg and Schroth, 2017;Lenstra et al., 2018).

**Author contribution**

SH and EK conceived and designed the study. SH carried out the fieldwork and lab work. SH and PP performed the XAS analyses and subsequent data treatment. SH, PP and EK analyzed the data. SH wrote the manuscript with support from PP and EK.

**Competing interests**

The authors declare that they have no conflict of interest.

**Acknowledgments**

Synchrotron work was done at beamline I811, MAX-lab synchrotron radiation source, Lund University, Sweden and at the high-brilliance X-ray absorption and X-ray emission spectroscopy undulator beamline ID26 of the European Synchrotron Radiation Facility (ESRF, Grenoble). We would like to thank Dr. Stefan Carlson for his support on site at the beamline I811.



Further, a special thanks to Dr. César Nicolás Cuevas for assistances on the analysis of the XAS data. We thank the Swedish University of Agricultural Sciences (SLU) and Swedish Meteorological and Hydrological Institute (SMHI) for monitoring data of Swedish river. Many thanks to all participants of the COCOA cruise in the Öre estuary in April 2015. A special thanks to Daniel Conley for his efforts in the COCOA project and proofreading of the manuscript. Thanks to Sofia Mebrahtu Wisén for analyzing the samples at the inorganic analytical laboratory. Financial support for this project was provided by the

Swedish Research Council (grant number 2015-05450), the Swedish Research council Formas through the strong research environment Managing the Multiple Stressors of the Baltic Sea (grant number 207-2010-126). Further, this study is a contribution of the BONUS project COCOA (grant agreement 2112932-1) funded jointly by the European Commission and FORMAS.

The supporting information contains: High resolution WT modulus of EXAFS data river Emån, Alster and Ljungby not

shown in the manuscript (Figure S1). $k^3$-weighted EXAFS spectra and Fourier transformations of all samples and corresponding model fits of all samples (Figure S2). Fe remaining in suspension in response of increasing salinity of *in situ* samples along a transect and theoretical for river Öre and Örekil (Figure S3). Table S1 showing the $k^3$-weighted Fe K-edge EXAFS fit results for all samples and Table S2, containing the EXAFS LCF results for the river mouth samples.

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





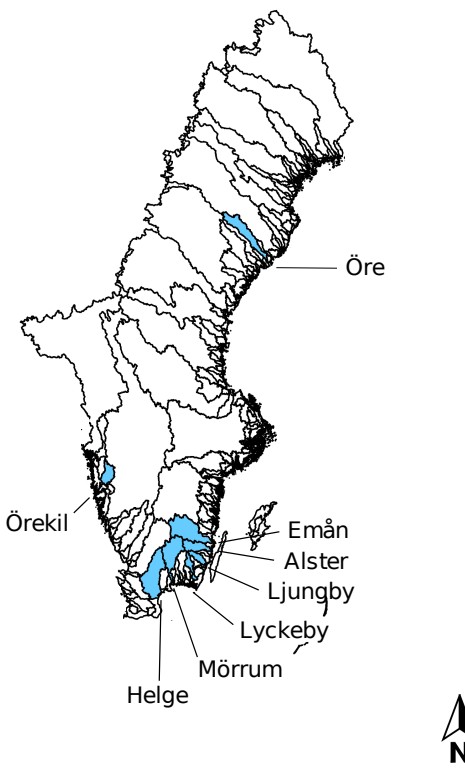

**Figure 1: Map of all river catchments in Sweden with the ones considered in this study named and marked blue.**




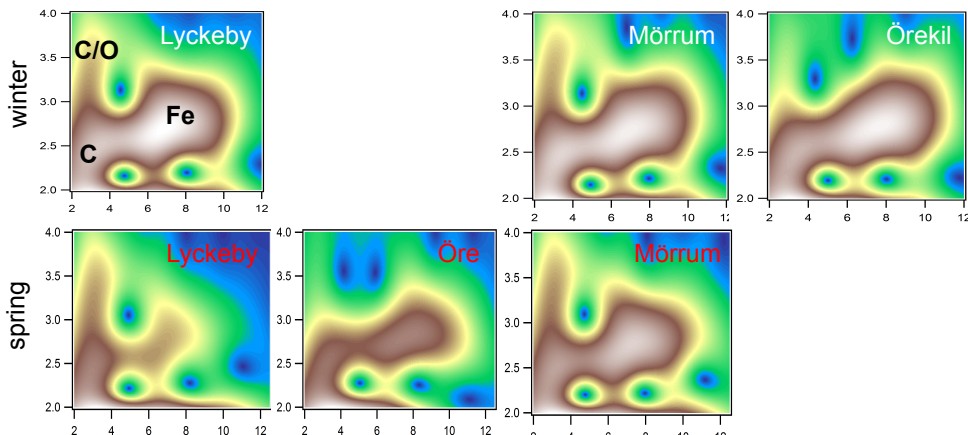

**Figure 2: Morlet wavelet transforms (η= 4, σ= 2) of EXAFS data collected on samples of river Lyckeby, Mörrum, Örekil and Öre (white=autumn, red spring) and are plotted as a function of k (Å⁻¹) on the x-axis and R (Å) on the y-axis. In the top left plot (Lyckeby) areas representing the different Fe scattering paths are indicated by C (Fe−C), C/O (Fe−C−C/O), and Fe (Fe−Fe).**



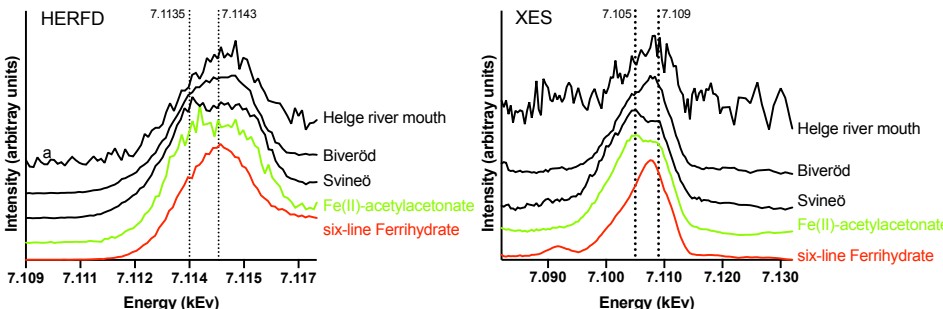

**Figure 3: Fe K-edge HERFD pre-edge spectra (left) and Kb$_{2,5}$ X-ray emission (XES) spectra (right) of the Helge river mouth sample, the Biveröd sample, the Svineö sample, Fe(II)-acetylacetonate (Fe(acac)$_2$) (green), and six-line Ferrihydrite (red). The spectra were normalized to the maximum intensities. The dotted lines are included for visual guidance.**



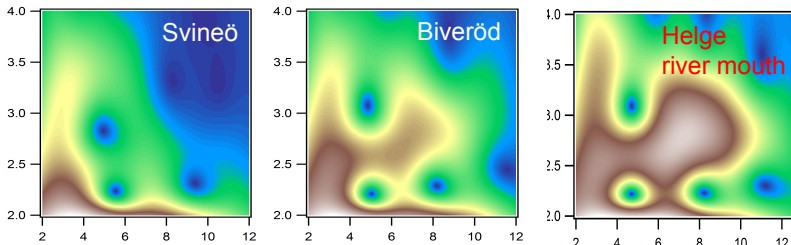

**Figure 4: Morlet wavelet transforms (η= 4, σ= 2) of EXAFS data of the two upstream samples Svineö and Biveröd and river mouth of the Helge catchment (white=autumn, red spring), plotted as a function of k (Å-1) on the x-axis and R (Å) on the y-axis.**




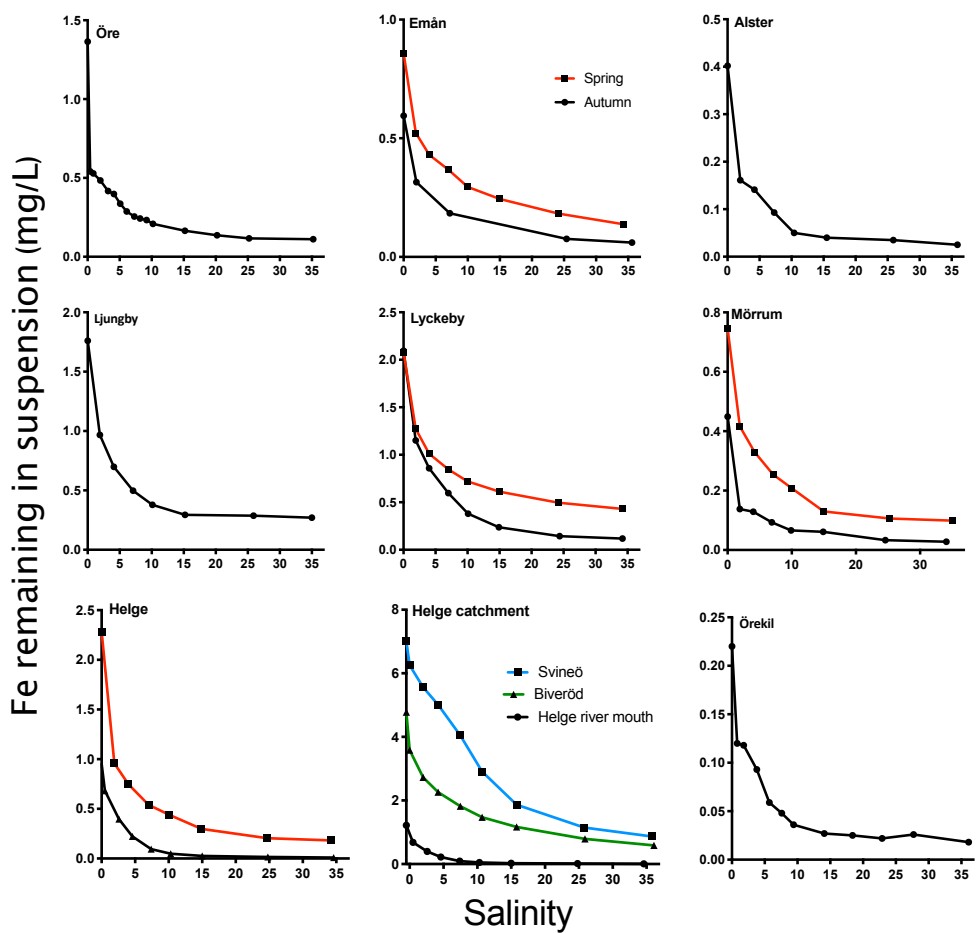

**Figure 5: The change in Fe in suspension in response to increasing salinity (0-35). Black lines denote sampling in autumn and red lines denote sampling in spring. For river Helge, the blue line denotes the most upstream sample (Svineö), the green line the other upstream sample (Biveröd) and the black line the river mouth.**






**Table 1: Catchment characteristics.**

| Site | Catchment area[a] | River length[a] | Discharge[a] (autumn/ spring) | Water retention time[b] | Forest cover[a] | Water cover[a] | Peat soil[a] |
|------|------------------|-----------------|-------------------------------|-------------------------|-----------------|----------------|--------------|
| | km² | km | m³/s | year | % | % | % |
| River Öre | 3029 | 225 | 8.2 | 0.5 | 71.8 | 3.1 | 25.3 |
| River Emån | 4471 | 220 | 24.1/65.4 | 1.4 | 73.3 | 6.1 | 8.4 |
| River Alster | 1525 | 100 | 4.5 | 1.0 | 79.7 | 5.1 | 9.6 |
| River Ljungby | 758 | 62 | 2.6 | 0.2 | 71.9 | 0.6 | 8.5 |
| River Lyckeby | 810 | 90 | 6.4/13.3 | 0.6 | 75.0 | 4.2 | 7.5 |
| River Mörrum | 3369 | 175 | 26.6/47.6 | 2.0 | 69.9 | 12.7 | 9.2 |
| Svineö* | 28 | | 0 | | 83.5 | 1.01 | 33.8 |
| Biveröd* | 44 | | 1 | | 92.6 | 0.9 | 11.6 |
| River Helge | 4724 | 190 | 14.9/36.9 | 0.5 | 57.5 | 4.8 | 14.4 |
| River Örekil | 1340 | 70 | 3.9 | 0.4 | 53.0 | 3.9 | 11.1 |

*Upstream sites in the Helge catchment. [a]Data obtained from http://vattenweb.smhi.se.[b]Data from Lindström et al. (2018).





**Table 2: Water chemistry and transport capacity at salinity 35 (corresponding to the salinity of the open sea) of collected samples.**

| Site | Sampling Date | pH | O$_2$ | Total Fe | DOC | Transport capacity at 35 salinity | CN ratio[a] | LCF ratio[b] |
|---|---|---|---|---|---|---|---|---|
| | | | % | mg/l | mg/l | % | | |
| River Öre | 20.04.2015 | 7.45 | - | 1.365 | 10.6 | 9.5 | 0.5 | 0.54 |
| River Emån | 03.11.2014 | 7.36 | 98 | 0.595 | 11.3 | 10.2 | 1.4 | 0.38 |
| | 09.03.2015 | 7.24 | - | 0.857 | 12.8 | 18.7 | 1.6 | 0.50 |
| River Alster | 03.11.2014 | 7.11 | 87 | 0.402 | 9.8 | 7.3 | 1.0 | 0.31 |
| River Ljungby | 20.10.2014 | 7.01 | 118 | 1.76 | 24.2 | 17.9 | 1.0 | 0.64 |
| River Lyckeby | 29.10.2014 | 6.99 | 101 | 2.095 | 19.4 | 6.6 | 1.0 | 0.50 |
| | 09.03.2015 | 6.55 | - | 2.082 | 19.4 | 24.1 | 2.7 | 0.89 |
| River Mörrum | 29.10.2014 | 7.43 | 103 | 0.449 | 10.2 | 7.3 | 1.5 | 0.53 |
| | 23.03.2015 | 7.05 | 105 | 0.745 | 13.6 | 15.5 | 2.0 | 0.71 |
| Svineö* | 06.11.2014 | 4.44 | 41 | 7.011 | 49.3 | 12.4 | - | 1.00 |
| Biveröd* | 06.11.2014 | 6.38 | 84 | 4.777 | 29.6 | 12.3 | 1.7 | - |
| River Helge | 06.10.2014 | 7.58 | 86 | 1.22 | - | 0.7 | | - |
| | 23.03.2015 | 7.47 | 101 | 2.280 | 15.8 | 9.3 | 1.6 | 0.46 |
| River Örekil | 14.07.2014 | 7.28 | 85 | 0.220 | 8.8 | 9.5 | 0.8 | 0.47 |

* Upstream sites in the Helge catchment. [a]Ratio of the coordination numbers of the fitting results, between the Fe-C path and the shortest (edge-sharing) Fe-Fe path (i.e. CN$_{Fe-C}$/CN$_{Fe-Fe}$). [b]Ratio of the Fe-OM fraction and the sum of Fe-oxide fractions from the LCF analysis.