# Peer review of "Organic Iron Complexes Enhance Iron Transport Capacity along Estuarine Salinity Gradients of Baltic Estuaries"

_Biogeosciences, 2019_

## Referee Comment (RC1) · Anonymous Referee #1 · 30 Jul 2019

Review of Organic Iron Complexes Enhance Iron Transport Capacity along Estuarine Salinity Gradients by Simon David Herzog, Per Persson, Kristina Kvashnina and Emma Sofia Kritzberg. Briefly, the manuscript presents the use of X-ray spectroscopy for the characterization of iron species in freezed-dried samples collected in a range of Scandinavian rivers ending in the Baltic Sea (except one in the Skagerrak). The study is complemented with mixing experiments with artificial seawater to imitate natural estuarine processes. The research group has a broad experience in the use of this specific technique and has published a series of interesting paper on iron speciation in Baltic rivers. The paper is a fine piece of work and brings forward many interesting conclusions about iron transition in riverine waters of boreal rivers. I really wish the authors would extend their area of study and produce similar works in rivers from other areas

covering catchments of different characteristics. I also think the authors should start collaborations with other groups that could provide other analytical techniques due to the limitations of the analytical approach shown in their work for species quantification. Overall, this is a very interesting work that brings a lot of qualitative information about the wide range of iron speciation that can be found in fairly similar estuaries. The manuscript supports the recent hypothesis that iron can scape high latitude estuaries in a percentage substantially higher than previously thought. The data has good quality and increases our understanding of estuarine processes. I recommend its publication after moderate revision mainly based on the need to increase the revision of literature (especially of literature referred to other analytical techniques) and improvable description of Fe changes under the increase of salinity. Sampling handling should be polished but I do not think this invalidates the manuscript. I have a few major concerns (nothing cannot be fixed): Data cooking is quite complicated and in principle very difficult to evaluate for an external reviewer and takes a few arguable assumptions. In principle, I have no doubts about the skills of the authors about this process. However, the result is a series of values without any indication of uncertainty or variability. This is a major issue in this kind of studies and in the few number of their papers I went through to prepare this review, there is no indication/description/estimation of the uncertainty associated to the values presented. I would appreciate a paragraph where the topic is addressed for a non initiated in the use of X-ray spectroscopy. So the reader can have an idea of the confidence can be given to the numbers presented in tables. It is also clear that trends obtained with different variables match but it is not clear how pecentaged of org/inor compare using different ratios. My second concern refers to the description of iron estuarine processes during the discussion. At the end of the discussion there is an approach to the real complexity of processes but in the first 3 sections there are over-simplifications. Example: "complexed Fe (Fe-OM) can "survive" the salinity gradient, while Fe (oxy)hydroxides are prone to aggregation and selectively removed". No possibility of FeO(OH) remaining in solution, OM is always described as a unique species where the real case is an extraordinary heterogeneity, ternary FeO(OH)-OM are only

considered to the end of discussion, fulvic vs humic , OM flocculation and precipitation, etc. I wonder how ternary phases FeO(OH)-OM would show in the WT contour plots. Would those separate on its components or create a third patch? Third, the use of artificial seawater for mixing experiments. This is an interesting experiment here to see the effect of ions but it is expected that marine OM plays a role in all these processes. So no surprise that the empirical transport parts ways from the "theoretical" value. The experiments presented here are perfectly valid and offer interesting results but the possibility that marine OM plays a role is not considered in the discussion. Fourth. There is a lot of literature not considered in the introduction. We have information about the fate of iron ligands in estuarine waters from recent work with voltammetric techniques (Buck, Lohan et al. 2007, Laglera and van den Berg 2009, Bundy, Abdulla et al. 2015, Su, Yang et al. 2015, Su, Yang et al. 2016, Yang, Su et al. 2017, Su, Yang et al. 2018). Fe transport is not specifically calculated but can be inferred from data. It is not about citing them all but at least acknowledging their existence and the hypotheses included. There is also interesting mixing work done on iron transport capacity with isotopic labelled iron (Krachler group) (Krachler, Jirsa et al. 2005, Krachler, Krachler et al. 2015) that is not referred. The manuscript relays too much in X-ray and partitioning techniques and does not cite the existence of other analytical approaches; it should include them in my opinion. There is also a recent paper that is much on the direction of this paper where it is determined the concentration of iron specifically bound to humics inclluding a profile of humic-rich Arctic waters (Sukekava, Downes et al. 2018).

Specific and minor comments:

Title: the title is generic and seems to be referring to global processes. The authors made a good effort sampling many rivers and repeating samplings in different seasons. The problem is that all rivers are from a small geographical region and refer to similar catchments, have similar conditions and end in the same regional sea. This is related to my opinion that the authors should use this interesting analytical approach in rivers from other locations. I think that the title should conceal the relevance of the study to

the area where it can be applied. I suggest "Organic Iron Complexes Enhance Iron Transport Capacity along Salinity Gradients of Baltic Estuaries". Fe speciation is not clearly defined as Fe transport is. Fe speciation sometimes refers to organic vs inorganic species and sometimes relates to the oxidation state of iron. My advice is to use organic speciation (or perhaps overall speciation) when org vs inorg is discussed and redox speciation when the Fe(II)/Fe(III) is discussed. How different noise in Fig 3 spectra translates in uncertainties at the time to report: example Helge river Abstract Lines 16-17. Example of oversimplification. All FeO(OH) precipitates and Fe-OM survives. Please add "a fraction" of the organically complexed . . . . . . . . ... I never heard of a study that suggest that all organic Fe "survives" the estuarine transit. Low-order stream? Not sure what is the meaning of the expression. Introduction: 28 iron is the fourth most abundant element in the earth crust and mantle, on earth it is possibly the most abundant element. 29 systemS. Please give range. I am not sure whether the authors refer here to freshwater systems or fresh and seawaters. 33 please add reference (Liu and Millero 2002). This is the main work on iron solubility in seawater and this paper is dedicated mostly to seawater. The Lofts paper is dedicated to freshwater. This paper should be the reference to discuss solubility 38 "suggesting that Fe export from soils are increasing." Check grammar, export is increasing

40 the Fe requirement in coastal waters is high but since it is not limiting I do not know if it plays a key biological role 48 "much higher than generally observed". This phrasing accepts several interpretations. I think the authors mean higher than predicted from prior works (the reference to 95% precipitation)

50 "Fe in natural waters is known to occur in two main phases, mononuclear organic complexes (Fe-OM) and Fe rich Fe (oxy)hydroxide colloids associated with chromophoric organic matter (Breitbarth et al., 2010;Hassellöv et al., 1999;Andersson et al., 2006)." In my opinion this is oversimplification, as it reads it seems that CDOM only can be found associated to Fe(OH)O and not forming soluble complexes. Electrochemical measurements have proved that fulvic and humic components of CDOM

bind iron forming complexes that can migrate freely to the electrode, i.e.: mononuclear (van den Berg and Laglera works on humics). This phrasing also assumes that associated to FeO(OH) there are no non coloured substances. This interpretation comes from the cited paper that include a peculiar description of iron complexation in rivers: "iron transport in rivers is associated with two types of carrier phases (besides detrital particles), an oxyhydroxide phase with associated CDOM (chromophoric dissolved organic matter, mostly consisting of humic acids) and an organic carbon (fulvic) phase (e.g. Lyv'en et al., 2003; Andersson et al., 2006)." From Breitbarth etal 2010). There is a huge body of literature that proves that fulvics belong to the CDOM fraction of DOM. Actually, the oversimplification that is found in the first 3 sections of this manuscript is diluted in the discussion section and there is a recount of the real complexity of the problem. Please rewrite this paragraph. 56 found "in" aggregates

Materials and methods 83 I am going to give a piece of advice to the authors for future work. Do not sample by hand in estuaries unless you have a system to open the bottle once it is at depth. Dipping an open bottle across the surface opens the possibility to collect a lot of surfactants and floating debris from the surface microlayer. I saw samplings ruined by this strategy. Fair paragraph at 90. I am interested in the concentration factor and should be included. Line 90 is a bit tricky because speciation here is obtained from the physical and not chemical properties of the sample (centrifugation=size partitioning prior to analysis). The warning is interesting because later on the manuscript the authors try to argue about the form of FeO(OH) crystallization. 94 how was the pH meter calibrated? TRIS or NBS solutions? Section 2.2. No problems with the approach. However I strongly recommend that for future studies they either obtain cleaner reagents or consider to remove metals from their working solutions. 150 nM Fe is a huge contamination and could interfere with some of the mixing experiments (obviously, the speciation of this contamination is different to the speciation of the sample). Please convert the blank concentration to mg Fe /L since this is the unit used throughout the paper. 110 24 h is a good compromise but has to be put into context. From (Liu and Millero 2002) work on Fe solubility in seawater "In our first series of measurements, we examined the solubility as a function of time. The results of iron solubility over 4 days at pH 3 and 8 are shown in Fig. 1. The iron concentrations decreased significantly from 3 h to 1 day and continued to decrease. After 1 week, the solubility did not decrease significantly. These results agree with our work in NaCl (Liu and Millero, 1999) and Kuma et al. (1996) in seawater. The subsequent solubility measurements were conducted with an equilibration time of at least 1 week. Our results represent the quasi-equilibration with iron solubility with particle size greater than 0.02 mm. Such a definition is in accordance with those of Byrne and Kester (1976a) and Kuma et al. (1996, 1998a,b). It may take several years for the solid to reach equilibrium (Schindler et al., 1963). The causes for the decrease in the solubility of Fe(III) with time has been recently been examined in more detail by Kuma et al. (1992, 1996, 1998a,b)." Please make a back of the envelop calculation about the fraction that is not removed in 24 h (I agree should be small) and cite Liu and Millero work. Also consider that with longer equilibrium times (as those residence times verified in estuaries, FeO(OH) aging could shift crystallization.

111 why 3000 rcf for 8 h?????? there is no bibliography attached as justification. What is approximately the size you discriminate? 2.3 Was any reference material analysed? Or at least in a previous work using exactly the same analytical settings? This is always required for oceanic studies. 118 problem with measuring pH with an electrode with a single calibration at changeable ionic strength. Again, how was this electrode calibrated? 2.4 and 2.5 I congratulate the authors for the degree of detail used to describe the analysis and data treatment steps. Again, the only thing I missed is a rough estimation of the uncertainty associated to the technique (specially the complex data treatment of the signals). 125 recorded? 140 Here I have a question. What is he concentration of Fe(III) added per mg of SRFA? Several recent reports state that the actual complexation is in the order of 15 nmol Fe per mg SRFA (Yang, Su et al. 2017, Slagter, Laglera et al. 2019). It could be that if iron was added greatly in excess, it was partially chemically bound and partially attached with other type of weaker interaction. I think a short explanation is important and future work could help to elucidate these

binding capacities reported by voltammetric methods. 175 please rewrite "were close to saturated with dissolved oxygen (85 – 118%)". 178 was this higher pH caused by biological or geochemical processes? 180 oxygen saturation suspiciously low. Was the temperature effect properly accounted for? This paper would greatly benefit of some sort of visual library (supplementary file?). Not clear from the text whether ferrihidrate and goethite show exactly the same contours 190 what compounds? Humics or mononuclear ligands? 193-198. Not sure tables S1 and S2 are correctly cited. Not in order for sure 208- it is clear there is a good correlation between the CNFe-C/CNFe-Fe and LCF ratios but my question is, are values comparable? Please add an statistic or visual comparison (for instance, if values of two different analytical approaches to the same parameter are close, the correlation should have slope close to one and Y-axis close to zero). 227- Are the authors referring to the Helge river here? In Helge pH is not low and O2 is not low 233-238 this paragraph is very difficult to evaluate without a rough idea of the uncertainty of the approach. Since the technique clearly struggles with quantification (although fantastic for qualitative analysis of multiple species) I would recommend for future work the combination with other speciation techniques. It could be that correlations have been hindered by the low number of data and uncertainty. 3.3 243 high removal but here the point would be, the remaining concentration is high or close to seawater concentrations in the Baltic sea (not reported here)? 243-244 This is very surprising and requires further discussion. Fe precipitation is usually the consequence of coprecipitation with organic phases after flocculation of organic matter (mostly humics) due to the increase of the ionic strength that cancels the negative charge of organic matter at natural pH (addition: this is discussed further in the discussion section but still. . ..). This is something described since Sholkovitz/Boyle papers. Therefore, the lack of OC precipitation is very surprising and not a result that mimics natural conditions. Here the authors need to elaborate much more in this result. If the experimental procedure somehow precluded the flocculation/coagulation of organic matter, then the precipitation of Fe was severely underestimated. This is for me the most worrying result in the manuscript. 251-252. This should have not been

done like this. Seawater has a minimum pH of 7.8 (function of course of temperature and salinity and local biochemical conditions); the perform of dilution experiments with pH at the high saline end member as low as 7.1 is not realistic. The authors should had forced the pH to realistic values ∼8. This could have modulated the precipitation of species during mixing experiments. 4. 265 I would rephrase to "in contrast to the thermodynamic modeling suggested by (Wällstedt et al., 2010) for these systems that predicted a dominance of ferrihydrite (≥97 %).". This prediction is subjective and depends in the parameters fed to the SHM model. I think that other research group could had obtained different results with the same model. Actually, I could not find the DOM concentration used in that specific paper.

268 I had concerns about the preservation of the redox speciation during sample processing but the result of Figure 3 is very revealing of the power of the analytical approach here. Kudos to the authors. Again, I would recommend for future work support from other ex-situ techniques more suited for quantification (spectrophotometry or chemiluminescence). 284 I do not think Sholkovitz was ever supportive of reduced Fe aggregation in any type of estuary independently of its latitude. The Powell paper shows complete iron precipitation (Figure 1). The idea of effective iron transport off high latitude rivers and humic rich streams was first put forward (to my knowledge) by Krachler and coauthors and it is deeply discussed in a recent review (Muller 2018). 285 "DOC was little affected by increasing salinity as previously observed (Linkhorst et al., 2017;Herzog et al., 2017)." This assertion is against prior observation by Sholkovitz, Boyle and other authors (I referred to this before), please discuss this finding and discrepancies among authors if exist or refer to the type of estuaries were this specific behaviour was observed. 289 these authors do not argue that the whole iron complexed to organic matter survives the estuarine transition. It is a bit more subtle although not against findings in this paper. The assertion is that against prior reports that sustained that all Fe coprecipitates with OM, a significant percentage of iron bound to DOM (in some works they specify to humic substances) "survives" estuarine mixing. Laglera and van den Berg argue that coprecipitation takes place down to a Fe/humics ratio

when both stabilize (or "learn to survive" if we continue with the metaphor). I advise rephrasing this section correcting the interpretation of prior literature and putting it into context with findings in this work. 294-295. First I would remove Sander's reference since this is a description of interactions at pH 4 under complete protonation of carboxylic groups. At pH 7-8 negative charges are dominant. 296-298. This is not exactly the common description of the estuarine transition of DOM and their interaction with inorganic iron. As cations increase and neutralize the surface groups of DOM, repulsion forces decrease and DOM starts flocculation. Many non charged colloids (such as Fe colloids) get trapped during this formation of bigger aggregations and coprecipitate eventually. Basically, the result is the same described in the paper but the authors suggest independent precipitation and the literature is full with text about combined precipitation. Actually, FeCl3 addition for organic matter coprecipitation and removal is a common procedure used in water treatment plants. 306 this assertions ignores a whole body of literature. Iron speciation at the concentrations found at the saline end member of estuarine is available after cathodic voltammetric methods (Stan van den Berg, Kristen Buck, Loes Gerringa, Han Su and many others). With those methods it is possible to measure the iron ligand concentrations and concentration of humic substances. It is true that it is not clear whether those methods may discern between stable Fe oxyhidroxydes and Fe-OM complexes but the reported ligand concentrations in excess of iron concentration can only be ascribed to the presence of organic ligands. It would be fair to do a short summary of findings and add to the discussion that organic ligands in excess of iron concentrations have been found by this technique. 310-315 there is a factor not considered. In the estuary, there is production of iron ligands by biota that could be used to explain why dissolved Fe in the estuary was higher than the predicted after experimentation with ligand free seawater. 323-324. Please add reference to Liu and Millero 2002. Some of the observations here could be easily predicted. 340. there could be other Fe(II) sources. For instance, at higher flow conditions probably there is more turbidity and less light penetration limiting Fe(II) photoproduction. Let alone biological production of Fe(II). The subject is very

complex. 360 which amounts? 370 this sentence has to be toned bown. First, this study is carried out in rivers which impact is never going to reach iron limited areas. Second, the number of iron limited areas in high latitudes of the northern hemisphere is not so extended (areas of the Bering Sea and perhaps after bloom in the Northern Atlantic). Third, the two studies referenced mention Arctic rivers, which are completely different catchment areas to those presented here since those are affected by permafrost melting. Not because there are iron limited areas and Fe from Baltic rivers is expected to increase, the Arctic Ocean is going to be fertilized. 377. I would remove last sentence. Although possible, it is very speculative and brings the focus out the main topic of the paper. Is there a first column missing in table S1? Buck, K. N., M. C. Lohan, C. J. M. Berger and K. W. Bruland (2007). "Dissolved iron speciation in two distinct river plumes and an estuary: Implications for riverine iron supply." Limnology and Oceanography 52(2): 843-855. Bundy, R. M., H. A. N. Abdulla, P. G. Hatcher, D. V. Biller, K. N. Buck and K. A. Barbeau (2015). "Iron-binding ligands and humic substances in the San Francisco Bay estuary and estuarine-influenced shelf regions of coastal California." Marine Chemistry 173(0): 183-194. Krachler, R., F. Jirsa and S. Ayromlou (2005). "Factors influencing the dissolved iron input by river water to the open ocean." Biogeosciences 2(4): 311-315. Krachler, R., R. F. Krachler, G. Wallner, S. Hann, M. Laux, M. F. Cervantes Recalde, F. Jirsa, E. Neubauer, F. von der Kammer, T. Hofmann and B. K. Keppler (2015). "River-derived humic substances as iron chelators in seawater." Marine Chemistry 174: 85-93. Laglera, L. M. and C. M. G. van den Berg (2009). "Evidence for geochemical control of iron by humic substances in seawater." Limnology and Oceanography 54(2): 610-619. Liu, X. W. and F. J. Millero (2002). "The solubility of iron in seawater." Marine Chemistry 77(1): 43-54. Muller, F. L. L. (2018). "Exploring the Potential Role of Terrestrially Derived Humic Substances in the Marine Biogeochemistry of Iron." Frontiers in Earth Science 6(159). Slagter, H. A., L. M. Laglera, C. Sukekava and L. J. A. Gerringa (2019). "Fe-binding Organic Ligands in the Humic-Rich TransPolar Drift in the Surface Arctic Ocean using Multiple Voltammetric Methods." Journal of Geophysical Research: Oceans 124: 1491-1508. Su, H.,

R. Yang, Y. Li and X. Wang (2018). "Influence of humic substances on iron distribution in the East China Sea." Chemosphere 204: 450-462. Su, H., R. Yang, I. Pižeta, D. Omanović, S. Wang and Y. Li (2016). "Distribution and Speciation of Dissolved Iron in Jiaozhou Bay (Yellow Sea, China)." Frontiers in Marine Science 3(99). Su, H., R. Yang, A. Zhang and Y. Li (2015). "Dissolved iron distribution and organic complexation in the coastal waters of the East China Sea." Marine Chemistry 173(0): 208-221. Sukekava, C., J. Downes, H. A. Slagter, L. J. A. Gerringa and L. M. Laglera (2018). "Determination of the contribution of humic substances to iron complexation in seawater by catalytic cathodic stripping voltammetry." Talanta 189: 359-364. Yang, R., H. Su, S. Qu and X. Wang (2017). "Capacity of humic substances to complex with iron at different salinities in the Yangtze River estuary and East China Sea." Scientific Reports 7(1): 1381.

---

## Referee Comment (RC2) · Anonymous Referee #2 · 26 Aug 2019

The authors present new data characterizing iron speciation in Scandinavian rivers together with Fe stability experiments aiming at estimating Fe transport across the salinity gradient to reach oceanic waters. While the work about Fe speciation seems rather well described and of high quality (for a non-specialist like I am), the work about Fe transport across the salinity gradient deserves more attention in my opinion. In addition, the authors seems to excessively generalize their findings. For instance the first sentence of the abstract is about 'open marine waters', while the most saline sample analyzed here has a salinity of 25 (seawater has a salinity of 35). Moreover, most studied rivers (7 out of 8) flow into the Baltic sea (typical salinities of 5 to 10) that is not proper seawater. Finally, the manuscript really lacks quantification (the authors state that fluxes could be 'significant' but no quantification is provided). The topic is

extremely interesting. I recommend publication in Biogeosciences only after the points below have been addressed.

Major points

1-Excessive generalization of results obtained mainly along the Baltic Sea. Authors should make clear from the title and abstract (and discussion and conclusion) that their study is regional, mainly along a sea with especially low salinity, and based on lab experiments (for the transport capacity).

2-Lack of quantification of the potential Fe source the authors talk about (L 23 'potentially bioavailable Fe' from rivers) compared to other Fe sources to the surface ocean. The authors should provide estimations of the different Fe sources to the ocean, so that the reader can make an opinion about the significance of the source discussed in the present paper compared to other sources. This is necessary to support for instance the 2 following sentences (L13-14 and L 23-24 below). - 'Rivers discharge a notable amount of Fe ($1.5x10^9$ mol yr $-1$ ) to coastal waters, but are still not considered important sources of bioavailable Fe to open marine waters' - 'This study suggests that boreal rivers may provide significant amounts of potentially bioavailable Fe to marine waters beyond the estuary, due to organic matter complexes.' The authors should remove assertions such as 'Fe loading from boreal rivers to estuaries is increasing substantially [. . .] this is a finding with major implications' (L 35 - 40) if they cannot present data showing that river dissolved Fe stabilized by organic ligands is indeed a significant flux compared to others for the surface ocean.

3-The core of the paper, in my opinion, reside in the fact that 2 main characteristics are studied, 1) Fe speciation and 2) Fe transport capacity, and that these 2 characteristics are compared to each other. However, while the first point, Fe speciation is well described in the ms (notably with 3 figures), the transport capacity experiment is hardly presented in the main part of the ms (data are almost only shown in the supplementary materials), so that the reader cannot really make an idea about the validity of the

author assertions. This is really a problem, because all the work about speciation is much less useful (at least in the presented context), if the transport capacity experiments are not validated. I believe that much more attention should be given to this part of the paper, with a proper discussion about the validity of the experiments, especially using the in situ data. In the main part of the ms, not in the supplement. Unfortunately, from what is shown in the supplement, I am not convinced that the mixing experiments do simulate accurately what would happen in situ. My opinion in that this dataset is insufficient to validate the transport capacities illustrated in Fig. 5 for instance. At least the authors should try to estimate error bars on the transport capacities (Table 2) and on the concentrations presented in Fig. 5. They should also mention that organic matter of oceanic origin (not reproduced in the lab mixing experiment) may also take part to the process. In addition, I think that the comparison between the 2 characteristics (speciation, transport) is also not sufficiently presented and described. L 245-247 'For the river mouth samples, the Fe transport capacity at 35 salinity correlated positively with the Fe speciation ratios (CN Fe- 245 C /CN Fe-Fe : r = 0.675, p = 0.023; LCF ratio: 0.78, p = 0.005). Further, Fe transport capacity at 35 salinity were negatively correlated to pH (r = -0.730, p = 0.007)' and L 291-293 ' The positive correlation between the contribution of Fe-OM (as determined by XAS) and Fe transport capacity (determined in artificial mixing experiments) adds a direct support that organic complexation of Fe is enhancing the stability across salinity gradients.'. I think that if the authors could provide a graphical representation of these correlations, this would be much easier for the reader and more convincing.

Minor points

Throughout the ms, the Fe phase the authors are talking about is not always clear. For instance, L 14 'the vast majority of riverine Fe', it seems that this is about dissolved Fe, but it is not mentioned. What's about particulate Fe ? Same for L 12. '1.5x109 mol yr-1'. For what phase ? etc.

L 13-14. 'Rivers discharge a notable amount of Fe (1.5x10 9 mol yr $-1$ ) to coastal

waters, but are still not considered important sources of bioavailable Fe to open marine waters'. This is not totally true in my opinion, because, since papers such as Radic et al 2011 or Labatut et al 2014, remobilization of particulate iron river discharges is presented as a major source. This comment is related to the preceding one.

L47. 'fraction of riverine Fe remaining in suspension'. A discussion about the phases involved would help clarify the ms. what about colloids, very small particles etc.

L56 'aggregates'. Check English

L63. XAS. Define

L86. 'cold'. What temperature ?

L 128 'were according'. Check English

L283. FeTC. Define.

L 378. ' the increases in Fe discharge is also likely to alter e.g. P retention in coastal sediments'. Again, this assertion should be supported by quantification.

---

## Author Comment (AC1) · 4 Oct 2019

The response to the referees comments are structures as follows: (1) comments from referees, (2) *author's response and author's suggested changes in manuscript (italic).*

Response to comments by referee #1:

Briefly, the manuscript presents the use of X-ray spectroscopy for the characterization of iron species in freezed-dried samples collected in a range of Scandinavian rivers ending in the Baltic Sea (except one in the Skagerrak). The study is complemented with mixing experiments with artificial seawater to imitate natural estuarine processes. The research group has a broad experience in the use of this specific technique and has published a series of interesting paper on iron speciation in Baltic rivers. The paper is a fine piece of work and brings forward many interesting conclusions about iron transition in riverine waters of boreal rivers. I really wish the authors would extend their area of study and produce similar works in rivers from other areas covering catchments of different characteristics. I also think the authors should start collaborations with other groups that could provide other analytical techniques due to the limitations of the analytical approach shown in their work for species quantification. Overall, this is a very interesting work that brings a lot of qualitative information about the wide range of iron speciation that can be found in fairly similar estuaries. The manuscript supports the recent hypothesis that iron can scape high latitude estuaries in a percentage substantially higher than previously thought. The data has good quality and increases our understanding of estuarine processes. I recommend its publication after moderate revision mainly based on the need to increase the revision of literature (especially of literature referred to other analytical techniques) and improvable description of Fe changes under the increase of salinity. Sampling handling should be polished but I do not think this invalidates the manuscript.

*Thank you for these positive remarks and constructive and interesting suggestions for further research within this theme. We have included literature as suggested and polished the description of sampling handling. The detailed response to each comment is listed below (in italic).*

I have a few major concerns (nothing cannot be fixed):

1. Data cooking is quite complicated and in principle very difficult to evaluate for an external reviewer and takes a few arguable assumptions. In principle, I have no doubts about the skills of the authors about this process. However, the result is a series of values without any indication of uncertainty or variability. This is a major issue in this kind of studies and in the few number of their papers I went through to prepare this review, there is no indication/description/estimation of the uncertainty associated to the values presented. I would appreciate a paragraph where the topic is addressed for a non initiated in the use of X-ray spectroscopy. So the reader can have an idea of the confidence can be given to the numbers presented in tables. It is also clear that trends obtained with different variables match but it is not clear how pecentaged of org/inor compare using different ratios.

*As noted by the referee, the approach used in this study to analyse and present XAS data has been used in previous work (Karlsson and Persson 2012; Sundman, et al. 2014). The two data evaluation techniques used in this study (CN- and LCF-ratios) are based on different modelling approaches of the XAS data to show the contribution of organic matter to the Fe*

*phases. The CN-ratio is based on the analysis of the X-ray absorption fine structure (EXAFS) region, providing information about the local coordination environment of Fe by quantitative modeling with input structures related to the natural samples. The LCF-ratio is based on a linear combination fitting (LCF) analysis by using a set of reference spectra, allowing to estimate the proportion of the two dominating Fe phases (Fe-OM complexes and Fe (oxy)hydroxides). There was agreement between the two ratios, i.e. a significant correlation between the CN- and the LCF-ratio was observed.*

*While XAS data is informative to the local structure around the selected element, in this case Fe, it is not strictly quantitative because of the large uncertainty in fitting the amplitude of the spectra, which mainly contain information about the contribution of each component. We therefore prefer to use the ratios to identify trends in the relative contribution of Fe-OM vs Fe (oxy)hydroxides and refrain from presenting exact percentages. For clarification the following has been added to the manuscript: "The XAS data contains information on the local structure around the selected element (Fe). It it is not strictly quantitative. Therefore, the ratios were merely used to identify trends in the relative contribution of Fe-OM vs Fe (oxy)hydroxides among the samples."*

*The confidence limits on the obtained distances and coordination numbers were estimated by Sundman, et al. (2013) by a procedure recommended by the International XAFS Society Standards and Criteria Committee (Sayers 2000). Each parameter was varied in a stepwise fashion away from its optimal value, while varying all other parameters until $\Delta\chi2$ increased 1.0 above its minimum value. This resulted in the following confidence limits: CN Fe–O: ±0.6; CN Fe–C: ±0.7; CN Fe–Fe: ±0.6; R Fe–O: ±0.01; R Fe–C: ±0.07; and R Fe–Fe: ±0.06. As we used the same experimental setup and modeling approach as Sundman et al. 2013, the same confidence limits apply to our data. The confidence limits have been added to the caption in Table S1, as follows: "Confidence limits on the obtained distances and coordination numbers estimated by a procedure recommended by the International XAFS Society Standards and Criteria Committee (Sayers 2000) performed by (Sundman, et al. 2013) CN Fe–O: ±0.6; CN Fe–C: ±0.7; CN Fe–Fe: ±0.6; R Fe–O: ±0.01; R Fe–C: ±0.07; and R Fe–Fe: ±0.06."*

2. My second concern refers to the description of iron estuarine processes during the discussion. At the end of the discussion there is an approach to the real complexity of processes but in the first 3 sections there are oversimplifications. Example: "complexed Fe (Fe-OM) can "survive" the salinity gradient, while Fe (oxy)hydroxides are prone to aggregation and selectively removed". No possibility of FeO(OH) remaining in solution, OM is always described as a unique species where the real case is an extraordinary heterogeneity, ternary FeO(OH)-OM are only considered to the end of discussion, fulvic vs humic , OM flocculation and precipitation, etc. I wonder how ternary phases FeO(OH)-OM would show in the WT contour plots. Would those separate on its components or create a third patch?

*The XAS technique captures local structures and this means that the method assigns Fe into either organically complexed or Fe(oxy)hydroxides. While it is very likely that molecules/colloids/particles in the suspension include ternary phases, the method does not distinguish this.*

3. Third, the use of artificial seawater for mixing experiments. This is an interesting experiment here to see the effect of ions but it is expected that marine OM plays a role in all these processes. So no surprise that the empirical transport parts ways from the "theoretical" value. The experiments presented here are perfectly valid and offer interesting results but the possibility that marine OM plays a role is not considered in the discussion.

*We will expand the discussion on the validity of the artificial seawater mixing experiments and bring in the comparison between the in situ and theoretical values of Fe concentration along salinity gradient into the manuscript, which was presented in the supplementary information. In this context we will also acknowledge the possible role of marine organic*

*matter. The following text addition is suggested to better describe the strengths and weaknesses of the artificial mixing experiments:" Results regarding Fe transport capacity derived from the artificial seawater mixing experiments were in good agreement with the estuarine transects sampled. Theoretically calculated Fe concentrations, based on Fe loss in artificial seawater mixing experiments with river water and the dilution factor, showed only minor deviations from Fe concentrations measured in the Gullmar Fjord. For the Öre estuary on the other hand, measured Fe concentrations were somewhat higher than the theoretical calculations (Figure S3). In the low-salinity mixing regime present in the northern Baltic (Bothnian Bay), aggregation may occur without significant sedimentation (Forsgren and Jansson 1992). This has been observed in the plume of nearby River Kalix, and was hypothesized to result from a high organic component of the aggregates, where low specific density may lead to transport of these aggregates far away from the river mouth (Gustafsson, et al. 2000). Thus, the centrifugation used to efficiently separate aggregates in the mixing experiments, may overestimate estuarine particle loss in this context. Despite the agreement between measured and theoretically estimated Fe concentrations, the artificial mixing experiments are unlikely to capture all processes that affect the loss of Fe along the natural salinity gradient. In the estuary, photoreduction may affect Fe speciation and affect its fate, as well as the occurrence of ligands produced by marine biota which may also influence the behaviour of riverine Fe. Indeed, the artificial mixing experiments capture the response of riverine Fe to increasing salinity in isolation, and how that depends on Fe speciation."*

4. Fourth, there is a lot of literature not considered in the introduction. We have information about the fate of iron ligands in estuarine waters from recent work with voltammetric techniques (Buck, Lohan et al. 2007, Laglera and van den Berg 2009, Bundy, Abdulla et al. 2015, Su, Yang et al. 2015, Su, Yang et al. 2016, Yang, Su et al. 2017, Su, Yang et al. 2018). Fe transport is not specifically calculated but can be inferred from data. It is not about citing them all but at least acknowledging their existence and the hypotheses included. There is also interesting mixing work done on iron transport capacity with isotopic labelled iron (Krachler group) (Krachler, Jirsa et al. 2005, Krachler, Krachler et al. 2015) that is not referred. The manuscript relays too much in X-ray and partitioning techniques and does not cite the existence of other analytical approaches; it should include them in my opinion. There is also a recent paper that is much on the direction of this paper where it is determined the concentration of iron specifically bound to humics inclluding a profile of humic-rich Arctic waters (Sukekava, Downes et al. 2018).

*The introduction includes studies using other analytical approaches, including papers by the Krachler group (Krachler et al., 2005, Krachler et al., 2010) based on isotopically labelled Fe, and papers using FIFFF (Hassellöv et al., 1999;Andersson et al., 2006; Stolpe and Hasellöv 2007). However referee#1 is right no studies based on voltammetric techniques are currently present in the introduction. The following addition to the introduction has been made: "Further, studies using cathodic stripping voltammetry (CSV) have underlined the importance of complexation by ligands to keep Fe in suspension in saline waters (Laglera and van den Berg 2009; Sukekava, et al. 2018).*

Specific and minor comments:

5. Title: the title is generic and seems to be referring to global processes. The authors made a good effort sampling many rivers and repeating samplings in different seasons. The problem is that all rivers are from a small geographical region and refer to similar catchments, have similar conditions and end in the same regional sea. This is related to my opinion that the authors should use this interesting analytical approach in rivers from other locations. I think that the title should conceal the relevance of the study to the area where it can be applied. I suggest "Organic Iron Complexes Enhance Iron Transport Capacity along Salinity Gradients of Baltic Estuaries".

*Suggestion implemented:"Organic Iron Complexes Enhance Iron Transport Capacity along Estuarine Salinity Gradients of Baltic Estuaries"*

6. Fe speciation is not clearly defined as Fe transport is. Fe speciation sometimes refers to organic vs inorganic species and sometimes relates to the oxidation state of iron. My advice is to use organic speciation (or perhaps overall speciation) when org vs inorg is discussed and redox speciation when the Fe(II)/Fe(III) is discussed.

*In this paper we assess both what you refer to as organic speciation (organic vs. inorganic) (EXAFS) and redox speciation (HERFED data). We have now clarified that: "The Fe speciation, (organic speciation (organic vs. inorganic) and redox speciation, of all river samples was characterized by XAS."*

7. How different noise in Fig 3 spectra translates in uncertainties at the time to report: example Helge river

*We are not entirely sure how to understand this comment. The HERFED spectra and $Kb_{2,5}$ X-ray emission (XES) spectra in Figure 3 of the river mouth are noisy. This is a result of the low Fe concentration, which translates into higher uncertainties in determining peak positions and relative peak intensities. Nevertheless, the difference between the Helge river and the Fe(II) spectra remains evident. We have included this observation in the discussion: "Due to low Fe concentration there was more noise in the river mouth sample, but the deviation from the Fe(II) spectra was still clear."*

Abstract
8. Lines 16-17. Example of oversimplification. All FeO(OH) precipitates and Fe-OM survives. Please add "a fraction" of the organically complexed. I never heard of a study that suggest that all organic Fe "survives" the estuarine transit.

*The sentence is not saying that all Fe-OM is surviving the estuarine transit, but rather that Fe-OM **can** survive the salinity gradient, while Fe (oxy)hydroxides **are more prone** to aggregation.*

9. Low-order stream? Not sure what is the meaning of the expression.
*The wording was changed to "upstream" and reads as followed:*
*"We further found that that Fe-OM was more prevalent at high flow conditions in spring than at low flow conditions during autumn, and that Fe-OM was more dominant upstream in a catchment than at the river mouth."*

Introduction:
10. 28 iron is the fourth most abundant element in the earth crust and mantle, on earth it is possibly the most abundant element.
*Thank for spotting this mistake. We have corrected it.*

11. 29 systemS.
*Thank you for pointing out this mistake.*

12. Please give range. I am not sure whether the authors refer here to freshwater systems or fresh and seawaters.
*This statement is general to fresh- and marine waters and the exception to that is illustrated in the continuation of the sentence. Thus providing a range of Fe concentration in aquatic systems serves no purpose.*

13. 33 please add reference (Liu and Millero 2002). This is the main work on iron solubility in seawater and this paper is dedicated mostly to seawater. The Lofts paper is dedicated to freshwater. This paper should be the reference to discuss solubility
*The reference (Liu and Millero 2002) was added. It is indeed relevant to give a marine reference here as our study spans from freshwater to marine water conditions.*

14. 38 "suggesting that Fe export from soils are increasing." Check grammar, export is

increasing
*Thank you for spotting this mistake.*

15. 40 the Fe requirement in coastal waters is high but since it is not limiting I do not know if it plays a key biological role

*Fe influences the mobility, availability and biogeochemistry of numerous other elements, especially it can play a key role in affecting phosphorous availability not only by limiting primary production. Further, Fe limitation in the Baltic Sea has been suggested by several studies, to clarify this the following section has been added: "This would suggest that high and rising concentrations of Fe from boreal rivers (Kritzberg and Ekstrom, 2012;Björnerås et al., 2017) may indeed result in increasing export of bioavailable Fe to the Baltic Sea and open waters, where it may limit N-fixation and primary production (Stal et al., 1999;Stolte et al., 2006;Martin and Fitzwater, 1988).2*

16. 48 "much higher than generally observed". This phrasing accepts several interpretations. I think the authors mean higher than predicted from prior works (the reference to 95% precipitation)

*Thank your for this input this is a much clearer way to express it: "Fe transport capacity – the fraction of riverine Fe remaining in suspension at higher salinity – has been shown to vary widely and is in some instances much higher than predicted from prior works (Kritzberg et al., 2014;Krachler et al., 2005)."*

17. 50 "Fe in natural waters is known to occur in two main phases, mononuclear organic complexes (Fe-OM) and Fe rich Fe (oxy)hydroxide colloids associated with chromophoric organic matter (Breitbarth et al., 2010;Hasellöv et al., 1999;Andersson et al., 2006)." In my opinion this is oversimplification, as it reads it seems that CDOM only can be found associated to Fe(OH)O and not forming soluble complexes. Electrochemical measurements have proved that fulvic and humic components of CDOM bind iron forming complexes that can migrate freely to the electrode, i.e.: mononuclear (van den Berg and Laglera works on humics). This phrasing also assumes that associated to FeO(OH) there are no non coloured substances. This interpretation comes from the cited paper that include a peculiar description of iron complexation in rivers: "iron transport in rivers is associated with two types of carrier phases (besides detrital particles), an oxyhydroxide phase with associated CDOM (chromophoric dissolved organic matter, mostly consisting of humic acids) and an organic carbon (fulvic) phase (e.g. Lyv0en et al., 2003; Andersson et al., 2006)." From Breitbarth etal 2010). There is a huge body of literature that proves that fulvics belong to the CDOM fraction of DOM. Actually, the oversimplification that is found in the first 3 sections of this manuscript is diluted in the discussion section and there is a recount of the real complexity of the problem. Please rewrite this paragraph.

*Thank you for taking the time to explain why this phrasing was problematic. We entirely agree that there is oversimplification and a degree of sloppiness in how the characteristic of the organic components are described. Our paper is focusing on the iron and it's speciation more than targeting the characteristics of the organic matter. We have therefore removed chromophoric as a descriptor of the organic matter interacting with the Fe (oxy)hydroxides: "Thus, the riverine Fe source to marine waters may be underestimated, especially for boreal rivers, where high DOC concentrations can affect Fe speciation. Fe in natural waters is known to occur in two main phases, mononuclear organic complexes (Fe-OM) and Fe-rich Fe (oxy)hydroxide colloids associated with organic matter (Breitbarth et al., 2010;Hasellöv et al., 1999;Andersson et al., 2006)."*

18. 56 found "in" aggregates Materials and methods
*Thank you for pointing this out, it has been corrected.*

19. 83 I am going to give a piece of advice to the authors for future work. Do not sample by

hand in estuaries unless you have a system to open the bottle once it is at depth. Dipping an open bottle across the surface opens the possibility to collect a lot of surfactants and floating debris from the surface microlayer. I saw samplings ruined by this strategy.
*Thank you for this valuable advice, we will keep this in mind.*

20. Fair paragraph at 90. I am interested in the concentration factor and should be included. Line 90 is a bit tricky because speciation here is obtained from the physical and not chemical properties of the sample (centrifugation= size partitioning prior to analysis). The warning is interesting because later on the manuscript the authors try to argue about the form of FeO(OH) crystallization.
*We are not sure how to understand this comment. This part described how samples were treated that were later analyzed by XAS for Fe speciation of total Fe in the river samples. There was no separation prior to this.*

21. 94 how was the pH meter calibrated? TRIS or NBS solutions?
*The pH meter was calibrated daily using a three point calibration with three buffer solutions (pH 4.01/7.00/10.00) purchased together with pH meter by Mettler Toledo. The pH calibration was not adjusted to the changing ionic strength. This may have an impact on the pH reading of the high salinity samples, however it should not have an affect on the freshwater samples on which our discussion builds. For further work we will take this valuable input into consideration when working with changing ionic strength.*

Section 2.2.
22. No problems with the approach. However I strongly recommend that for future studies they either obtain cleaner reagents or consider to remove metals from their working solutions. 150 nM Fe is a huge contamination and could interfere with some of the mixing experiments (obviously, the speciation of this contamination is different to the speciation of the sample).
*Thank you for this comment, we will consider this for our future work. Since this contamination is between 0.1-3.8% of the Fe concentration in our river waters, we do not think this has affected our results.*

23. Please convert the blank concentration to mg Fe /L since this is the unit used throughout the paper.
*Thank you for this input the blank (0.15 μM) has been converted into mg/l (0.0025 mg/l) as it is used in this unit throughout the paper.*

24. 110 24 h is a good compromise but has to be put into context. From (Liu and Millero 2002) work on Fe solubility in seawater "In our first series of measurements, we examined the solubility as a function of time. The results of iron solubility over 4 days at pH 3 and 8 are shown in Fig. 1. The iron concentrations decreased significantly from 3 h to 1 day and continued to decrease. After 1 week, the solubility did not decrease significantly. These results agree with our work in NaCl (Liu and Millero, 1999) and Kuma et al. (1996) in seawater. The subsequent solubility measurements were conducted with an equilibration time of at least 1 week. Our results represent the quasi-equilibration with iron solubility with particle size greater than 0.02 mm. Such a definition is in accordance with those of Byrne and Kester (1976a) and Kuma et al. (1996, 1998a,b). It may take several years for the solid to reach equilibrium (Schindler et al., 1963). The causes for the decrease in the solubility of Fe(III) with time has been recently been examined in more detail by Kuma et al. (1992, 1996, 1998a,b)." Please make a back of the envelop calculation about the fraction that is not removed in 24 h (I agree should be small) and cite Liu and Millero work. Also consider that with longer equilibrium times (as those residence times verified in estuaries, FeO(OH) aging could shift crystallization.
*Salinity induced aggregation of Fe consists of sequential reactions. Nowostawska, et al. (2008) showed that a significant fraction of Fe (~80 %) is aggregated immediately within a*

*few seconds after mixing river water and sea salt solution. Furthermore, Hunter and Leonard (1988) demonstrated that aggregation of riverine Fe after the addition of sea salt is well described second-order kinetics where the rate of aggregation decline with time. The figure below (Figure 1) exemplifies how a clear increase in aggregation of riverine Fe was observed for the first ~100 minutes but additional aggregation after 2 hours was minor. Thus, while aggregation is continuing at a slow rate also after 24 hours, the first 24 hours should incorporate the dominant part of the aggregation.*

[Figure]

Figure 1 from *Hunter and Leonard (1988)*

25. 111 why 3000 rcf for 8 h?????? there is no bibliography attached as justification. What is approximately the size you discriminate?

*A similar protocol has been used in previously published work by Krachler et al 2005 and by our group (Herzog et al. 2019, Kritzberg et al. 2014) and as well. We have no estimate of the size discriminated, bur from other work we see that filtration through 0.2 μm- filters remova a large fraction of reactive Fe that remains stable in the water column. Thus. Separating by size may not be the best way to reflect loss from the water column. Centrifugation acts on gravitational particles, which may be of different size and density.*

26. Was any reference material analysed? Or at least in a previous work using exactly the same analytical settings? This is always required for oceanic studies.

*We don't know to what analytical method the referee is referring too. If it was DOC, the same analytical setting has been used in previous work from our group and many others.*

27. 118 problem with measuring pH with an electrode with a single calibration at changeable ionic strength. Again, how was this electrode calibrated?

*This has already been addressed (Comment 21)*

28. 2.4 and 2.5 I congratulate the authors for the degree of detail used to describe the analysis and data treatment steps. Again, the only thing I missed is a rough estimation of the uncertainty associated to the technique (specially the complex data treatment of the signals).

*Thank you, the estimation of the uncertainty associated to the technique has been addressed in response to comment 1.*

29. 125 recorded?

*Thank you for pointing this out, it has been adjusted.*

30. 140 Here I have a question. What is he concentration of Fe(III) added per mg of SRFA?

Several recent reports state that the actual complexation is in the order of 15 nmol Fe per mg SRFA (Yang, Su et al. 2017, Slagter, Laglera et al. 2019). It could be that if iron was added greatly in excess, it was partially chemically bound and partially attached with other type of weaker interaction. I think a short explanation is important and future work could help to elucidate these binding capacities reported by voltammetric methods.

*The Fe(III) concentrations in Suwannee River natural organic matter for the reference material used in this study was 6489 µg g$^{-1}$ on a dry mass basis, used in Karlsson and Persson (2012). For clarification the two techniques (voltammetric methods and XAS) measure different aspects to distinguish Fe speciation. The basic principles behind XAS is that X-rays strike and excite core electrons of an atom, which in turn get either promoted to an unoccupied level, or ejected from the atom and consequently create a core hole. Dissociation of the electron will produce an excited ion as well as a photoelectron. The scattering of the photoelectron will modulate the absorption coefficient, and both the local transitions and the effects from the out-going photoelectron can be measured and analyzed. As XAS provides information on the local coordination environment of Fe, the concern by refree#1 about the what is measured (chemically bound or partially attached with other type of weaker interaction) is not of concerns*

31. 175 please rewrite "were close to saturated with dissolved oxygen (85 – 118%)".
*Thank for pointing this out, the sentence has been corrected as follow: "At the time of sampling all river mouths were close to saturation with dissolved oxygen (85 – 118%) and pH values close to neutral (Table 2)."*

32. 178 was this higher pH caused by biological or geochemical processes?
*The higher pH was the result of the low level of DOC measured during autumn. This is further discussed in the manuscript in section 4.3 (Control of spatial variation and flow conditions on Fe speciation.).*

33. 180 oxygen saturation suspiciously low. Was the temperature effect properly accounted for?
*The oxygen saturation at the upstream sample (Svineö) is correct and the temperature effect was accounted for as the OxyGuard probes have built-in temperature compensation. Low oxygen is often found in waters that drain peatbogs, since the organic matter degradation in the standing water is consuming oxygen.*

34. *This paper would greatly benefit of some sort of visual library (supplementary file?). Not clear from the text whether ferrihidrate and goethite show exactly the same contours.*
*We assume this comment refers to the section 3.2 (XAS characterization). The main purpose of XAS characterization was to distinguish between organically complexed Fe and Fe (oxy)hydroxides and it was not the goal to completely resolve the structure of the Fe (oxyhydroxide fraction. When we talk about Fe (oxy)hydroxides, we compare it to reference material from both ferrihydrite and goethite as both have a similar Fe-bond distances.*
*To make the XAS results easier to understand we added WT plots of the model compounds ferrihydrite, goethite and two plots showing a mixture of goethite and trisoxalatoiron(III) to the supplement (Figure S2) and referred to them in section 3.2. We hope these additions will make the XAS analyses and interpretations easier to grasp.*

*Addition to the Supplement:*
*For the quantitative modeling of the EXAFS spectra two input structures were used, goethite for the Fe-Fe paths and trisoxalatoiron(III) for the Fe-O, Fe-C and Fe-C/O. WT plots for the model compounds (Figure S2), show that the different paths in the model compounds are in good agreement with the ones found in our samples (Figure 2). The Fe (oxy)hydroxides ferrihydrite and goethite (Figure S2 a and b) are represented in the same area of the WT plots, so that distinction between the two is difficult.*

[Figure]

Figure 2 Figure S2. Morlet wavelet transforms for a (η = 10, σ = 1) and for b-c (η = 8, σ = 1) of EXAFS data displaying the second and third coordination shells collected on model compounds: (a) 6-line ferrihydrite by Sundman et al. (2014), (b) goethite, (c) goethite/trisoxalatoiron(III) mixture (50:50), and (d) goethite/trisoxalatoiron(III) mixture (25:75) by Karlsson and Persson (2010).

35. 190 what compounds? Humics or mononuclear ligands?
*As our approach was aiming to distinguish between the two Fe phases - organically complexed Fe and Fe (oxy)hydroxide – the purpose of this section was to point out the difference between these two with WT plots. We are not characterizing the organic matter.*

36. 193-198. Not sure tables S1 and S2 are correctly cited. Not in order for sure
*Thank you for pointing this out, the reference to Table S2 on line 194 was wrong and it should have been referring to Table S1, this has been corrected.*

37. 208- it is clear there is a good correlation between the CNFe-C/CNFe-Fe and LCF ratios but my question is, are values comparable? Please add an statistic or visual comparison (for instance, if values of two different analytical approaches to the same parameter are close, the correlation should have slope close to one and Y-axis close to zero).
*The CN- and LCF-ratios are obtained by two different data evaluation techniques and are two different ways measuring the contribution of organic matter in the Fe phases. As requested by referee #1 a visual comparison (see comment 1) has been added into the supplementary.*

38. 227- Are the authors referring to the Helge river here? In Helge pH is not low and O2 is not low
*Thank you for pointing this out. In this section we are referring to the upstream samples of Helge River, which was not entirely clear in the current sentence. It has been adjusted as follows: "Finally, comparing the various EXAFS analyses with the HERFED and Kb2,5 emission spectroscopy results show that Fe(II) in the upstream samples of the Helge river system is present as Fe-OM complexes. These complexes are favored by low pH and low oxygen concentrations, as expected."*

39. 233-238 this paragraph is very difficult to evaluate without a rough idea of the uncertainty of the approach. Since the technique clearly struggles with quantification (although fantastic for qualitative analysis of multiple species) I would recommend for future work the combination with other speciation techniques. It could be that correlations have been hindered by the low number of data and uncertainty.

*This comment has already been addressed (comment 1).*

40. 243 high removal but here the point would be, the remaining concentrationis high or close to seawater concentrations in the Baltic sea (not reported here)?
*The high riverine input of Fe and OM, in combination with the relatively low salinity, render Fe concentrations in the Baltic Sea high (15–144 nmol L-1; Bothian Sea –Baltic proper, (Gelting et al. 2010) compared to the open sea. The values obtained by the mixing experiment at high salinity (35) by not considering the dilution factor are much higher (0.4-9.0 µmol L-1)*

*compared to values measure in the Baltic Sea.*
*This information has been implemented into the manuscript, as also referee#2 has commended on a similar issues: "Nevertheless, the high values of Fe remaining in suspension due to complexation with organic matter at high salinity (0.02 mg/l – 0.50 mg/l) supports that the importance of rivers as a source of Fe into the Baltic Sea with an Fe concentration of around 1 μg/l (Baltic proper) (Gelting, et al. 2010)."*

41. 243-244 This is very surprising and requires further discussion. Fe precipitation is usually the consequence of coprecipitation with organic phases after flocculation of organic matter (mostly humics) due to the increase of the ionic strength that cancels the negative charge of organic matter at natural pH (addition: this is discussed further in the discussion section but still: : :.). This is something described since Sholkovitz/Boyle papers. Therefore, the lack of OC precipitation is very surprising and not a result that mimics natural conditions. Here the authors need to elaborate much more in this result. If the experimental procedure somehow precluded the flocculation/coagulation of organic matter, then the precipitation of Fe was severely underestimated. This is for me the most worrying result in the manuscript.

*The results presented in this study go along with previous studies from similar system within the Baltic Sea catchment (Herzog et al., 2017;Forsgren et al.(Kritzberg, et al. 2014), 1996). It is indeed very likely that there is co-precipitation with organic matter, but since the DOC pool is so large, this does not result in a significant loss. .*

42. 251-252. This should have not been done like this. Seawater has a minimum pH of 7.8 (function of course of temperature and salinity and local biochemical conditions); the perform of dilution experiments with pH at the high saline end member as low as 7.1 is not realistic. The authors should had forced the pH to realistic values _8. This could have modulated the precipitation of species during mixing experiments.

*It is correct that pH has an effect, since higher pH promotes colloid formation, due to increased Fe(III) hydrolysis and Fe(II) oxidation (Ilina, et al. 2013; Karlsson and Persson 2012; Pullin and Cabaniss 2003).However, such an effect is mainly important when moving within a lower pH range (3.0−6.7)(Karlsson and Persson 2012; Neubauer, et al. 2013) than we have in this study (>pH 7.1), and it has been previously verified that pH alone does not significantly affect Fe stability in the pH range of the current experiment (Kritzberg, et al. 2014).*

43. 265 I would rephrase to "in contrast to the thermodynamic modeling suggested by (Wällstedt et al., 2010) for these systems that predicted a dominance of ferrihydrite (_97 %).". This prediction is subjective and depends in the parameters fed to the SHM model. I think that other research group could had obtained different results with the same model. Actually, I could not find the DOM concentration used in that specific paper.

*The suggestion was implemented.*

44. 268 I had concerns about the preservation of the redox speciation during sample processing but the result of Figure 3 is very revealing of the power of the analytical approach here. Kudos to the authors. Again, I would recommend for future work support from other ex-situ techniques more suited for quantification (spectrophotometry or chemiluminescence).

*Thank you for this comment. We were also very pleased to see the high FeII measured in the upstream samples supporting that redox state is preserved. Supporting this with other techniques is a good suggestion.*

45. 284 I do not think Sholkovitz was ever supportive of reduced Fe aggregation in any type of estuary independently of its latitude. The Powell paper shows complete iron precipitation (Figure 1). The idea of effective iron transport off high latitude rivers and humic rich streams was first put forward (to my knowledge) by Krachler and coauthors

and it is deeply discussed in a recent review (Muller 2018).
*Thank you for this input, the references has been changes accordingly:* "*The high Fe transport capacity for most of those Swedish rivers studied go along with the existing literature showing that high-latitude DOC-rich rivers exhibit higher Fe-carrying capacities (Krachler et al., 2005;Muller, 2018).*"

46. 285"DOC was little affected by increasing salinity as previously observed (Linkhorst et al., 2017;Herzog et al., 2017)." This assertion is against prior observation by Sholkovitz, Boyle and other authors (I referred to this before), please discuss this finding and discrepancies among authors if exist or refer to the type of estuaries were this specific behavior was observed.
*The sentence has been adjusted and reads now as follows:*
"*DOC was little affected by increasing salinity as previously observed in such high latitude rivers with high DOC concentrations (Herzog et al., 2017;Forsgren et al., 1996).*"

47. 289 these authors do not argue that the whole iron complexed to organic matter survives the estuarine transition. It is a bit more subtle although not against findings in this paper. The assertion is that against prior reports that sustained that all Fe coprecipitates with OM, a significant percentage of iron bound to DOM (in some works they specify to humic substances) "survives" estuarine mixing. Laglera and van den Berg argue that coprecipitation takes place down to a Fe/humics ratio when both stabilize (or "learn to survive" if we continue with the metaphor). I advise rephrasing this section correcting the interpretation of prior literature and putting it into context with findings in this work.
*The section has been changed as follows:*
*In contrast, Fe complexed by terrigenous organic matter is supposedly less affected and to a larger extent "surviving" estuarine mixing and can thereby be a source of bioavailable Fe to marine waters (Batchelli, et al. 2010; Krachler, et al. 2010; Laglera and van den Berg 2009).*

48. 294-295. First I would remove Sander's reference since this is a description of interactions at pH 4 under complete protonation of carboxylic groups. At pH 7-8 negative charges are dominant.
*The reference has been removed.*

49. 296-298. This is not exactly the common description of the estuarine transition of DOM and their interaction with inorganic iron. As cations increase and neutralize the surface groups of DOM, repulsion forces decrease and DOM starts flocculation. Many non charged colloids (such as Fe colloids) get trapped during this formation of bigger aggregations and coprecipitate eventually. Basically, the result is the same described in the paper but the authors suggest independent precipitation and the literature is full with text about combined precipitation. Actually, $FeCl_3$ addition for organic matter coprecipitation and removal is a common procedure used in water treatment plants.
*We are not suggesting independent precipitation. The Fe (oxy) hydroxides lost are most likely in association with organic matter and "bring this down", it is only that that OM is only a minor fraction of the total OM. This is now clarified in the manuscript: "With increasing salinity, the surface charge gets neutralized resulting in reduced colloidal repulsion (Mosley, et al. 2003) and formation aggregates containing both Fe and OM."*

50. 306 this assertions ignores a whole body of literature. Iron speciation at the concentrations found at the saline end member of estuarine is available after cathodic voltammetric methods (Stan van den Berg, Kristen Buck, Loes Gerringa, Han Su and many others). With those methods it is possible to measure the iron ligand concentrations and concentration of humic substances. It is true that it is not clear whether those methods may discern between stable Fe oxyhidroxides and Fe-OM complexes but the reported ligand concentrations in excess of iron concentration can only be ascribed to the presence

of organic ligands. It would be fair to do a short summary of findings and add to the discussion that organic ligands in excess of iron concentrations have been found by this technique.

*Thank you for pointing this out. The following section was added:*
*"Moreover, based on cathodic stripping voltammetry (CSV), ligand concentrations have been found to be in excess of iron concentration, suggesting that organic ligands are complexing the Fe and keeping it in suspension in saline waters (Gledhill and Buck 2012; Laglera, et al. 2011)."*

51. 310-315 there is a factor not considered. In the estuary, there is production of iron ligands by biota that could be used to explain why dissolved Fe in the estuary was higher than the predicted after experimentation with ligand free seawater.

*Thank you for pointing this out. The production of ligands by the biota will be mentioned in an expanded text that addresses the validity of our artificial mixing experiments (see comment 3).*

52. 323-324. Please add reference to Liu and Millero 2002. Some of the observations here could be easily predicted.

*Thank you for this input, the reference was added.*

53. 340. there could be other Fe(II) sources. For instance, at higher flow conditions probably there is more turbidity and less light penetration limiting Fe(II) photoproduction. Let alone biological production of Fe(II). The subject is very complex.

*It is true that there are a different of sources and processes promoting Fe(II) in freshwater. This section is discussing the seasonal variation of the two main sources of Fe-OM and Fe (oxy)hydroxides into the rivers, especially the formation of Fe (oxy)hydroxide based on the rapid oxidation of Fe(II) from groundwater input. These systems have a minimum of turbidity and phytoplankton. Thus elaborate more on other may be relevant in other systems, however in the context it is not relevant.*

54. 360 which amounts?

*The word "amounts" has been exchanged with "quantity" to make it clearer that it is refers to Fe exported from the catchment. The sentence reads now accordingly:*
*"While characteristics such as land-cover and soil type are most likely affecting both quantity and speciation of Fe exported from the catchment, the limited number of rivers and sampling occasions of this study cannot accurately discern such relationships."*

55. 370 this sentence has to be toned bown. First, this study is carried out in rivers which impact is never going to reach iron limited areas. Second, the number of iron limited areas in high latitudes of the northern hemisphere is not so extended (areas of the Bering Sea and perhaps after bloom in the Northern Atlantic). Third, the two studies referenced mention Arctic rivers, which are completely different catchment areas to those presented here since those are affected by permafrost melting. Not because there are iron limited areas and Fe from Baltic rivers is expected to increase, the Arctic Ocean is going to be fertilized.

*Thank you for pointing this out. We have made the sentence specific to the Baltic only and refer to papers that indicate periods of Fe limitation in the Baltic.*
*In regard to the third point, two studies with rivers in the catchments of the Baltic Sea have been added, showing the same trends as the Arctic Rivers mentioned previously. The sentence reads now as follows: "Major hydrological events like spring floods and heavy storms have been observed to increase of the Fe concentration by up to a factor of 20 and alter the annual Fe load in northern rivers (Hölemann et al., 2005;Rember and Trefry, 2004;Dahlqvist et al., 2007;Herzog et al., 2019)."*

56. 377. I would remove last sentence. Although possible, it is very speculative and brings

the focus out the main topic of the paper.
*The sentence has been removed.*

57. Is there a first column missing in table S1?
*No there is no column missing. The table size was too large for the page format.*
*This has been changed and the table is now fully visible.*

**References from referee#1:**
Buck, K. N., M. C. Lohan, C. J. M. Berger and K. W. Bruland (2007). "Dissolved iron speciation in two distinct river plumes and an estuary: Implications for riverine iron supply." Limnology and Oceanography 52(2): 843-855. Bundy, R. M., H. A. N. Abdulla, P. G. Hatcher, D. V. Biller, K. N. Buck and K. A. Barbeau (2015). "Iron-binding ligands and humic substances in the San Francisco Bay estuary and estuarine-influenced shelf regions of coastal California." Marine Chemistry 173(0): 183-194. Krachler, R., F. Jirsa and S. Ayromlou (2005). "Factors influencing the dissolved iron input by river water to the open ocean." Biogeosciences 2(4): 311-315. Krachler, R., R. F. Krachler, G. Wallner, S. Hann, M. Laux, M. F. Cervantes Recalde, F. Jirsa, E. Neubauer, F. von der Kammer, T. Hofmann and B. K. Keppler (2015). "River-derived humic substances as iron chelators in seawater." Marine Chemistry 174: 85-93. Laglera, L. M. and C. M. G. van den Berg (2009). "Evidence for geochemical control of iron by humic substances in seawater." Limnology and Oceanography 54(2): 610-619. Liu, X. W. and F. J. Millero (2002). "The solubility of iron in seawater." Marine Chemistry 77(1): 43-54. Muller, F. L. L. (2018). "Exploring the Potential Role of Terrestrially Derived Humic Substances in the Marine Biogeochemistry of Iron." Frontiers in Earth Science 6(159). Slagter, H. A., L. M. Laglera, C. Sukekava and L. J. A. Gerringa (2019). "Fe-binding Organic Ligands in the Humic-Rich TransPolar Drift in the Surface Arctic Ocean using Multiple Voltammetric Methods." Journal of Geophysical Research: Oceans 124: 1491-1508. Su, H., R. Yang, Y. Li and X. Wang (2018). "Influence of humic substances on iron distribution in the East China Sea." Chemosphere 204: 450-462. Su, H., R. Yang, I. Pižeta, D. Omanoviĉc, S. Wang and Y. Li (2016). "Distribution and Speciation of Dissolved Iron in Jiaozhou Bay (Yellow Sea, China)." Frontiers in Marine Science 3(99). Su, H., R. Yang, A. Zhang and Y. Li (2015). "Dissolved iron distribution and organic complexation in the coastal waters of the East China Sea." Marine Chemistry 173(0): 208-221. Sukekava, C., J. Downes, H. A. Slagter, L. J. A. Gerringa and L. M. Laglera (2018). "Determination of the contribution of humic substances to iron complexation in seawater by catalytic cathodic stripping voltammetry." Talanta 189: 359-364. Yang, R., H. Su, S. Qu and X.Wang (2017). "Capacity of humic substances to complex with iron at different salinities in the Yangtze River estuary and East China Sea." Scientific Reports 7(1): 1381.

**References from the responds by the authors:**
Batchelli, Silvia, et al.
    2010   Evidence for strong but dynamic iron– humic colloidal associations in humic-rich coastal waters. Environmental science & technology 44(22):8485-8490.
Forsgren, Gunilla, and Mats Jansson
    1992   The turnover of river-transported iron, phosphorus and organic carbon in the Öre estuary, northern Sweden. *In* Sediment/Water Interactions. Pp. 585-596: Springer.
Gelting, J., et al.
    2010   Fractionation of iron species and iron isotopes in the Baltic Sea euphotic zone. Biogeosciences 7(8):2489-2508.
Gledhill, M., and K. N. Buck
    2012   The organic complexation of iron in the marine environment: a review. Front Microbiol 3:69.

Gustafsson, Örjan, et al.

    2000   Colloid dynamics and transport of major elements through a boreal river—brackish bay mixing zone. Marine Chemistry 71(1-2):1-21.

Hunter, Keith A, and Margaret W Leonard

    1988   Colloid stability and aggregation in estuaries: 1. Aggregation kinetics of riverine dissolved iron after mixing with seawater. Geochimica et Cosmochimica Acta 52(5):1123-1130.

Ilina, Svetlana M., et al.

    2013   Extreme iron isotope fractionation between colloids and particles of boreal and temperate organic-rich waters. Geochimica et Cosmochimica Acta 101:96-111.

Karlsson, Torbjörn, and Per Persson

    2012   Complexes with aquatic organic matter suppress hydrolysis and precipitation of Fe(III). Chemical Geology 322–323(0):19-27.

Krachler, R., et al.

    2010   Relevance of peat-draining rivers for the riverine input of dissolved iron into the ocean. Sci Total Environ 408(11):2402-8.

Kritzberg, Emma S., et al.

    2014   Importance of Boreal Rivers in Providing Iron to Marine Waters. PLoS ONE 9(9):e107500.

Laglera, Luis M, Gianluca Battaglia, and Constant MG van den Berg

    2011   Effect of humic substances on the iron speciation in natural waters by CLE/CSV. Marine Chemistry 127(1):134-143.

Laglera, Luis M, and Constant MG van den Berg

    2009   Evidence for geochemical control of iron by humic substances in seawater. Limnology and Oceanography 54(2):610-619.

Mosley, Luke M, Keith A Hunter, and William A Ducker

    2003   Forces between colloid particles in natural waters. Environmental science & technology 37(15):3303-3308.

Neubauer, E., et al.

    2013   Effect of pH and stream order on iron and arsenic speciation in boreal catchments. Environ Sci Technol 47(13):7120-8.

Pullin, Michael J, and Stephen E Cabaniss

    2003   The effects of pH, ionic strength, and iron–fulvic acid interactions on the kinetics of non-photochemical iron transformations. I. Iron (II) oxidation and iron (III) colloid formation. Geochimica et cosmochimica acta 67(21):4067-4077.

Sayers, DE

    2000   Report of the International XAFS Society Standards and Criteria Committee. the IXS Standards and Criteria Committee.

Sukekava, Camila, et al.

    2018   Determination of the contribution of humic substances to iron complexation in seawater by catalytic cathodic stripping voltammetry. Talanta 189:359-364.

Sundman, A., T. Karlsson, and P. Persson

    2013   An experimental protocol for structural characterization of Fe in dilute natural waters. Environ Sci Technol 47(15):8557-64.

Sundman, Anneli, et al.

2014   XAS study of iron speciation in soils and waters from a boreal catchment. Chemical Geology 364:93-102.

---

## Author Comment (AC2) · 4 Oct 2019

The response to the referees comments are structures as follows: (1) comments from referees, (2) *author's response and author's suggested changes in manuscript (italic).*

Response to comments by referee #2:

The authors present new data characterizing iron speciation in Scandinavian rivers together with Fe stability experiments aiming at estimating Fe transport across the salinity gradient to reach oceanic waters. While the work about Fe speciation seems rather well described and of high quality (for a non-specialist like I am), the work about Fe transport across the salinity gradient deserves more attention in my opinion. In addition, the authors seems to excessively generalize their findings. For instance the first sentence of the abstract is about 'open marine waters', while the most saline sample analyzed here has a salinity of 25 (seawater has a salinity of 35). Moreover, most studied rivers (7 out of 8) flow into the Baltic sea (typical salinities of 5 to 10) that is not proper seawater. Finally, the manuscript really lacks quantification (the authors state that fluxes could be 'significant' but no quantification is provided). The topic is extremely interesting. I recommend publication in Biogeosciences only after the points below have been addressed.

*Nice to hear that the topic is found to be interesting and that we are given the opportunity to address points raised. As the comments above are further elaborated by the referee in the list of major points, we respond and describe suggested changes to each specific comment below.*

Major points:
1. Excessive generalization of results obtained mainly along the Baltic Sea. Authors should make clear from the title and abstract (and discussion and conclusion) that their study is regional, mainly along a sea with especially low salinity, and based on lab experiments (for the transport capacity).

*Thanks for this comment. Our intention was not to suggest that our results can be generalized to all systems, but rather to put the topic in a general context. While the systems we work in are indeed atypical given the low salinity in the Baltic Sea, the general response of riverine Fe to increasing salinity is likely to be comparable in other regions. The response is probably more influenced by the water chemistry of the rivers than the salinity gradient, considering that the loss of Fe from suspension appear to occur at salinities below 15. Nevertheless, it is important that the results are not overstated and that the reader is not mislead about how far conclusions can be drawn. We have gone through the manuscript with this in mind and suggest the following changes:*
*In the abstract we clarify the geographical region in which the study is performed: "In this study, we directly identified, by X-ray absorption spectroscopy, the occurrence of these two Fe phases across **eight boreal rivers draining into the Baltic Sea**, and confirmed a significant but variable contribution of Fe-OM in relation to Fe (oxy)hydroxides among river mouths." Moreover, we removed the reference to marine waters in the concluding sentence of the abstract: "This study suggests that boreal rivers may provide significant amounts of potentially bioavailable Fe **beyond the estuary**, due to organic matter complexes." We also*

*clarify that Fe transport capacity was assessed by lab experiments: "The stability of Fe to increasing salinity, **as assessed by artificial mixing experiments**, correlated well to the relative contribution of Fe-OM, confirming that organic complexes promote Fe transport capacity."*
*In the introduction we also clarify the geographic region of the study and the fact that the rivers drain into the brackish Baltic Sea: "To this purpose, we sampled eight river **mouths that drain at the Swedish coast into the brackish Baltic Sea**."*
*The low salinity, particularly of the northern Baltic is now explicit in the Discussion "In the **low-salinity mixing regime present in the northern Baltic (Bothnian Bay),** aggregation may occur without significant sedimentation (Forsgren and Jansson, 1992)."*
*Finally in the conclusion: "This would suggest that high and rising concentrations of Fe from boreal rivers (Kritzberg and Ekstrom, 2012;Björnerås et al., 2017) may indeed result in increasing export of bioavailable Fe **to the Baltic Sea** and open waters, where it may limit N-fixation and primary production (Stal et al., 1999;Stolte et al., 2006;Martin and Fitzwater, 1988)."*

2. Lack of quantification of the potential Fe source the authors talk about (L 23 'potentially bioavailable Fe' from rivers) compared to other Fe sources to the surface ocean. The authors should provide estimations of the different Fe sources to the ocean, so that the reader can make an opinion about the significance of the source discussed in the present paper compared to other sources. This is necessary to support for in-stance the 2 following sentences (L13-14 and L 23-24 below). - 'Rivers discharge a notable amount of Fe (1.5x10 9 mol yr–1 ) to coastal waters, but are still not considered important sources of bioavailable Fe to open marine waters' - 'This study suggests that boreal rivers may provide significant amounts of potentially bioavailable Fe to marine waters beyond the estuary, due to organic matter complexes.' The authors should remove assertions such as 'Fe loading from boreal rivers to estuaries is increasing substantially [...] this is a finding with major implications' (L 35 - 40) if they cannot present data showing that river dissolved Fe stabilized by organic ligands is indeed a significant flux compared to others for the surface ocean.

*Thank you for this comment. In our view, the first sentence of the abstract is there to provide a general context. The elaboration on quantifying different sources as suggested by referee#2 is complex and would require more text than can be fitted into an abstract. Furthermore, the increasing Fe loading from boreal waters will likely have major implications also if not stabilized by organic ligands. Studies have shown that Fe of riverine origin is a phosphorus sink in coastal sediments, for instance. The reasoning is that what the specific implications may be depend on the fate of Fe across the salinity gradient.*

3. The core of the paper, in my opinion, reside in the fact that 2 main characteristics are studied, 1) Fe speciation and 2) Fe transport capacity, and that these 2 characteristics are compared to each other. However, while the first point, Fe speciation is well de-scribed in the ms (notably with 3 figures), the transport capacity experiment is hardly presented in the main part of the ms (data are almost only shown in the supplementary materials), so that the reader cannot really make an idea about the validity of the author assertions. This is really a problem, because all the work about speciation is much less useful (at least in the presented context), if the transport capacity experiments are not validated. I believe that much more attention should be given to this part of the paper, with a proper discussion about the validity of

the experiments, especially using the in situ data. In the main part of the ms, not in the supplement.

*In the original submission, the Fe transport capacity was presented in Figure 5 and Table 2 of the main manuscript, and the comparison of in situ Fe concentration along estuarine salinity gradients and theoretically estimated concentration based on the artificial salinity experiments were presented in Figure S3 in the supplementary information. In response to the above comment, we suggest to move the latter Figure into the main manuscript and expand the discussion on the validity of the artificial mixing experiments. For suggested text addition see response to comment 5 below.*

4. Unfortunately, from what is shown in the supplement, I am not convinced that the mixing experiments do simulate accurately what would happen in situ. My opinion in that this dataset is insufficient to validate the transport capacities illustrated in Fig. 5 for instance. At least the authors should try to estimate error bars on the transport capacities (Table 2) and on the concentrations presented in Fig. 5.

*It is important that mixing experiments are initiated as soon as possible after sampling, to make sure that Fe speciation is not altered. Moreover, the number of samples that can be included and processed in the experiment within a reasonable timeframe after sampling, is limited by centrifugation capacity. In the trade-off between running experimental replicates at a few selected salinity levels and including a wide gradient with many levels of salinity, we chose the latter, as we believe this provides more information. The consistency in the gradual loss of Fe in suspension with increasing salinity is in itself a validation of the Fe transport capacity measured at high salinity. We agree that the artificial mixing experiments are unlikely to capture exactly the loss of Fe along the natural salinity gradient, where for instance photoreduction may play a role, as well as the occurrence of organic matter of marine origin which may interact with riverine Fe and influence its behaviour. We also agree that these limitations should have been clearly recognized. Indeed, the experimental setup we apply capture the response of riverine Fe to increasing salinity in isolation. For suggested text addition see response to comment 5 below.*

5. They should also mention that organic matter of oceanic origin (not reproduced in the lab mixing experiment) may also take part to the process.

*We agree that this should be mentioned. The following text addition is suggested to better describe the strengths and weaknesses of the artificial mixing experiments:" Results regarding Fe transport capacity derived from the artificial seawater mixing experiments were in good agreement with the estuarine transects sampled. Theoretically calculated Fe concentrations, based on Fe loss in artificial seawater mixing experiments with river water and the dilution factor, showed only minor deviations from Fe concentrations measured in the Gullmar Fjord. For the Öre estuary on the other hand, measured Fe concentrations were somewhat higher than the theoretical calculations (Figure S3). In the low-salinity mixing regime present in the northern Baltic (Bothnian Bay), aggregation may occur without significant sedimentation (Forsgren and Jansson, 1992). This has been observed in the plume of nearby River Kalix, and was hypothesized to result from a high organic component of the aggregates, where low specific density may lead to transport of these aggregates far away from the river mouth (Gustafsson et al., 2000). Thus, the centrifugation used to efficiently separate aggregates in the mixing experiments, may overestimate estuarine particle loss in this context. Despite the agreement between measured and theoretically estimated Fe concentrations, the artificial mixing experiments are unlikely to capture all processes that affect the loss of Fe along the natural salinity gradient. In the estuary, photoreduction may affect Fe speciation and affect its fate, as well as the occurrence of ligands produced by marine biota which may also influence the behaviour of riverine Fe. Indeed, the artificial*

*mixing experiments capture the response of riverine Fe to increasing salinity in isolation, and how that depends on Fe speciation."*

6. In addition, I think that the comparison between the 2 characteristics (speciation, transport) is also not sufficiently presented and described. L 245-247 'For the river mouth samples, the Fe transport capacity at 35 salinity correlated positively with the Fe speciation ratios (CN Fe- 245 C /CN Fe-Fe : r = 0.675, p = 0.023; LCF ratio:0.78, p = 0.005). Further, Fe transport capacity at 35 salinity were negatively correlated to pH (r = -0.730, p = 0.007)' and L 291-293 ' The positive correlation between the contribution of Fe-OM (as determined by XAS) and Fe transport capacity (determined in artificial mixing experiments) adds a direct support that organic complexation of Fe is enhancing the stability across salinity gradients.'. I think that if the authors could provide a graphical representation of these correlations, this would be much easier for the reader and more convincing.

*Thanks for this input. To follow this advice, we suggest the addition of a figure to the supplementary that visually demonstrates the relationship between Fe transport capacity and the contribution of Fe-OM (as determined by XAS).*

[Figure]

Figure S5 Relationship between Fe transport capacity at 35 salinity and relative contribution of organically complexed Fe as assessed by the CN-ratio (A) and LCF-ratio (B).

**Minor points**

7. Throughout the ms, the Fe phase the authors are talking about is not always clear. For instance, L 14 'the vast majority of riverine Fe', it seems that this is about dissolved Fe, but it is not mentioned. What's about particulate Fe ? Same for L 12. '1.5x109 molyr-1'. For what phase ? etc. L 13-14. 'Rivers discharge a notable amount of Fe (1.5x10 9 mol yr−1 ) to coastal waters, but are still not considered important sources of bioavailable Fe to open marine waters'. This is not totally true in my opinion, because, since papers such as Radicet al 2011 or Labatut et al 2014, remobilization of particulate iron river discharges is presented as a major source. This comment is related to the preceding one.

*While size distributions are not a focus of this manuscript - organically complexed Fe and Fe (oxy)hydroxides are overlapping in size and can span from dissolved to particulate – we have gone over the manuscript to avoid unclarity as to the Fe phase referred to. Moreover, it is correct that iron that has settled to the sediment may be remobilized. While we cannot elaborate on this in the abstract, we have included this in the introduction: "Moreover, benthic release of Fe and subsequent lateral transport, was recently found to be a significant source of dissolved Fe to open marine waters (van Hulten et al., 2017)."*

8. L47. 'fraction of riverine Fe remaining in suspension'. A discussion about the phases involved would help clarify the ms. what about colloids, very small particles etc.

*This comment has already been address (see above).*

9. L56 'aggregates'. Check English

*Thanks – corrected.*

10. L63. XAS. Define

*Thank you for pointing this out the abbreviation will be written out: "The Fe speciation of all river samples was characterized by X-ray absorption spectroscopy (XAS)."*

11. L86. 'cold'. What temperature?

*The samples were stored in cooling boxes with freezing blocks during transport to keep the water from warming, This will be clarified: "Samples were stored cold and dark in a cooling box with freezing elements until return to the lab."*

12. L 128 'were according'. Check English

*The phrasing has been changed and reads now: "… was performed according to …"*

13. L283. FeTC. Define.

*Thank you for pointing this out, FeTC has been replaced by "Fe transport capacity"*

14. L 378. ' the increases in Fe discharge is also likely to alter e.g. P retention in coastalsediments'. Again, this assertion should be supported by quantification.

*The sentence has been removed based on a comment by referee #1 (Comment 56).*